# Cocoa, livelihoods, and deforestation within the Tridom landscape in the Congo Basin: A spatial analysis

Jonas Ngouhouo-Poufoun[1,2,3]* , Sabine Chaupain-Guillot[4], Youba Ndiaye[4‡], Denis Jean Sonwa[5‡], Kevin Yana Njabo[6‡], Philippe Delacote[4,7]

**1** Department of Geography, University College of London, London, United Kingdom, **2** International Institute of Tropical Agriculture (IITA), Yaoundé, Cameroon, **3** Congo Basin Institute (CBI), Yaoundé, Cameroon, **4** AgroParisTech, CNRS, INRAE, BETA, University of Lorraine, University of Strasbourg, Nancy, France, **5** Center for International Forestry Research, Jl. CIFOR-ICRAF, Yaoundé, Cameroon, **6** Center for Tropical Research, University of California, Los Angeles, Los Angeles, California, United States of America, **7** Climate Economics Chair, Paris, France

☉ These authors contributed equally to this work.
‡YD, DJS and KYN also contributed equally to this work.
* ngouhouo8p8j@gmail.com

**Data Availability Statement:** The data underlying this article are available in the CKAN Open Access repository for research data at https://doi.org/10.25502/w4tp-vn35/d.

## Abstract

In the context of emerging international trade regulations on deforestation-free commodities, the drivers of households' deforestation in conservation landscapes are of interest. The role of households' livelihood strategies including cocoa production, and the effects of human-elephant conflict are investigated. Using a unique dataset from a survey of 1035 households in the Tridom landscape in the Congo basin, the spatial autoregressive model shows that: (1) Households imitate the deforestation decisions of their neighbors; (2) A marginally higher income from cocoa production-based livelihood portfolios is associated with six to seven times higher deforestation compared to other livelihood strategies with a significant spillover effect on neighboring households' deforestation. The increase in income, mainly from cocoa production-based livelihoods in open-access systems can have a negative effect on forests. Households with a higher share of auto-consumption are associated with lower deforestation. If economic development brings better market access and lower auto-consumption shares, this is likely to positively influence deforestation. Without proper land use planning/zoning associated with incentives, promoting sustainable agriculture, such as complex cocoa agroforestry systems, may lead to forest degradation and deforestation.

## 1 Introduction

Globally about 75% of the poor population live in rural areas, with approximately 90% involved in farming as a way of earning a living [1–4]. Forest resources which account for about 22% to 27% of total households' income [5–8], also play an important role in terms of services, products, and incomes [9–12]. Forests are widely recognized as safety nets to mitigate agricultural risk, to help people cope with seasonal, climatic, and other stressors

**Funding:** We acknowledge funding for the Staff (JNP) time from the UK Research and Innovation's Global Challenges Research, Fund (UKRI GCRF) through the Trade, Development and the Environment Hub project (project number ES/S008160/1). https://gtr.ukri.org/projects?ref=ES%2FS008160%2F1. The Center for International Forestry Research-Global Comparative Study (CIFOR-GCS M3) has contributed to the fieldwork with funding provided by the Norwegian Agency for Development Cooperation (NORAD), grant no: QZA-12/0882. Grants received by JNP. https://www.norad.no/en/front/. The UMR BETA is supported by a grant overseen by the French National Research Agency (ANR) as part of the "Investissements d'Avenir" program (ANR-11-LABX-0002-01, Lab of Excellence ARBRE). https://anr.fr/en/. The funders had no role in study design, data collection, and analysis, decision to publish, or preparation of the manuscript.

**Competing interests:** The authors have declared that no competing interests exist.

[13, 14]. Rural households switch between specialization and diversification to optimize their livelihood provisioning [15–17]. In the Dja-Odzala-Minkébé tri-national transboundary conservation landscape (Tridom landscape) in the Congo basin, Ngouhouo-Poufoun et al. [98] investigated the variables determining the household choice to specialize or diversify its activities. Choosing a livelihood strategy in the Tridom landscape can be seen as a strategic choice between forest-based and non-forest-based or agriculture-based portfolios. The agricultural-based portfolio here includes small-scale farming and/or internationally traded commodities such as cocoa. Depending on the orientation between land-converting activities and forest resource extraction, effort allocation by households might either increase deforestation, increase forest degradation, or both [18]. Indeed, agricultural expansion to satisfy local, national, and international trade, drives almost 90% of global deforestation, contributing to 10 to 12% of the total global annual anthropogenic greenhouse gases (GHG) emissions [19–22].

The transformation from forest to agricultural land is threatening biodiversity conservation and causing GHG emissions. New analysis shows that just seven agricultural commodities (cattle, oil palm, soy, cocoa, rubber, coffee, and plantation wood fiber) accounted for 26% of global tree cover loss from 2001 to 2015, replacing 71.9 million hectares of forest during that period, an area of land more than twice the size of Germany [23]. The Guinean rain forest (GRF) of West Africa, identified over 30 years ago as a global biodiversity hotspot, had reduced to 11.3 million ha at the start of the new millennium that is 18% of its original area due to extensive smallholder agriculture [24]. From 1988 to 2007, the area deforested in the GRF by smallholders of cocoa, cassava, and oil palm increased by 6.8 million hectares (*Ibid., p. 307*). The ongoing expansion of cocoa farming has contributed to the loss of 80% of rainforest cover in some African countries [25].

The rural landscape in sub-Saharan Africa is made of a mixture of different land uses including food crops and agroforestry systems. In West Africa, it is now established that the promotion of unshaded cocoa has contributed to large-scale deforestation in countries such as Ivory Coast and Ghana [26, 27]. In West and Central Africa, including the Congo Basin, promoting a complex cocoa farming system that generally mimics forest structure contributes to forest degradation with less damage to natural resources [27, 28]. The type of farming systems and the way it is promoted can have a significant implication on forest local resources in rural landscapes in the forest fringe [26].

Inappropriate use of natural resources, poaching, and non-sustainable harvesting of non-timber forest products (NTFPs) can have significant adverse impacts on biodiversity and forest ecosystems, and lead to forest degradation [29–31, 81], reducing the capacity of the forest to regenerate and to produce ecosystem services [32, 33].

Our recent household surveys in the Tridom landscape scale provide some evidence that 85% of households are responsible for changing forest to other uses, regardless of their livelihood strategies. Population density in this landscape is low, less than 7 $inh./km^2$, and local households are less likely to practice optimal crop rotation. There is no binding regime of land acquisition in the non-permanent forest estate (NPFE), legally open for competing use including agriculture. The NPFE is seen by the communities as a common access resources, where the local people can clear relatively large areas of land at low cost [34]. A binding regime or more secure property rights motivates efficient resource management by landowners [35]. We observed during the field survey for complementary research in 2021 that some village chiefs are still offering newcomers in the villages large areas of land, often more than 50ha, without following the land acquisition and registration processes.

The forest sector in the Congo Basin Countries is divided into (i) 'Permanent Forest Estate', which includes logging concessions, ought to remain forest and mandated to maintain the

biological diversity, and (ii) 'Non-Permanent Forest Estate', that can be turned to alternative use including sustainable agriculture.

In many cases, extensive and unsustainable household farming based on slash-and-burn cultivation exacerbates small-scale deforestation and forest degradation. After selling the soft-wood lumber, all the remaining plants and materials in the forest are burnt and land is used for extensive agricultural production. In almost all cases, land conversion is done without revival of forest neither artificially nor naturally. Indeed, rural households are rarely involved in reforestation activities, while primary or secondary forest is progressively replaced by cork-wood, whose carbon storage potential is very low. Soil fertility and crop yields decline in the process [36, 37] and may cause a food crop production loss of at least $2.4billion to $5 billion across the Congo Basin [38]. Without a good fallow system, local people experience poor agricultural yields per hectare. Indeed, at least 75% of cocoa and plantain yields observed are less than 0,338t/ha and 3.59t/ha respectively with an average of 0.236t/ha and 3,09t/ha. This average yield is below the known average performance given limited farm means of production, which is 0.5t/ha and 16.5 t/ha, respectively. The potential yield of cocoa is 0.73t/ha and 1.22t/ha when cocoa plantations are associated with timber shade and leguminous tree species respectively. In the Talba cocoa production basin in Cameroon, potential yield can reach 1.6t/ha when the trees are between 10 and 20 years old [39]. When there is a good use of the litter fall, the maximum yield can reach 2.4t/ha [40, 41]. Regarding plantains, the potential yield can reach 30t/ha/year [42]. The diminishing returns due to unsustainable practices contribute to the perpetuation of poverty [43]. Hence, the high level of forest dependence may not necessarily correspond to a high and sustainable potential to reduce poverty [44]. Rather, this may lead to over-exploitation of common access resources and constitute a poverty trap when rural households face a large need for insurance [45].

While international trade in agricultural commodities such as cocoa and wildlife can spur economic development especially where governance is strong, there are also unavoidable social and environmental impacts [46, 47] and entails a higher risk of deforestation. The importance of sustainability in the agricultural system has never been more prominent than it is now [48].

To meet the diverse needs of both nature and communities, it is crucial to develop farming systems that prioritize sustainability. This includes addressing concerns such as deforestation, biodiversity conservation, climate change adaptation and mitigation, and improving overall productivity. One concept that has emerged in response to these challenges is Climate Smart Agriculture (CSA) [49]. Its adoption would lessen the effects of climate change in subsistence agriculture [50]. A well-managed, complex, cocoa agroforestry is seen as a sustainable tool for forest landscapes. This approach not only benefits local farmers but also contributes to the preservation of nature and the overall landscape [51]. New regulations are being developed internationally, aiming at decoupling commodities such as cocoa from deforestation and enhancing biodiversity. For example, we have the European Union's regulation on deforestation-free products, the European Union's Corporate Sustainability Due Diligence Directive (CSDDD), and the United Kingdom's due diligence law. These regulations aim to prevent commodities that are the products of illegal and legal deforestation and degraded ecosystems from coming into the EU and the UK markets by obliging in-scope businesses to conduct due diligence on their supply chains [52, 53].

In light of the above considerations, analyzing the full set of potential drivers of households' deforestation, prioritizing or distinguishing among them in order to inform policymakers and facilitate appropriate political decision processes to curb deforestation from smallholders' agriculture and forest activities in the medium and long term perspectives is of crucial interest [54, 55]. Our paper seeks to answer the following questions. How and how much do cocoa production and the different livelihood strategies developed by households, given wildlife

constraints such as human-wildlife conflict, impact small-scale deforestation? Deforestation here stands for any transformation of forest land covers to any agricultural land uses during the past decade to the survey without consideration of any legal deforestation defined within the REDD+ process. REDD+ stands for "Reducing emissions from deforestation and forest degradation, together with sustainable forest management, conservation, and enhancement of forest carbon stocks (REDD+)" It is a critical part of global efforts to mitigate climate change. FAO supports developing countries in their REDD+ processes but also helps them to translate their political commitments, as presented in their Nationally Determined Contributions, into action on the ground.

The following sections develop the literature review and our contribution (section 2), Objectives and hypothesis (section 3), and a simple microeconomic model (section 4). The spatial economic procedure is presented in section 5, the results in section 6 and discussion and conclusion in section 7.

## 2 Literature review and contribution

### 2.1 Literature review

Academic research on the causes of tropical deforestation relevant to this study includes (1) conceptual frameworks, (2) macro-level empirical studies including regional and national levels, (3) micro-level empirical studies, and (4) spatially explicit analyses.

**2.1.1 Conceptual framework related studies.**   The first analysis that combined the results of multiple studies to frame the causes of tropical deforestation was realized by Angelsen and Kaimowitz [56] They synthesized the results of more than 140 economic models using five types of variables to build a framework for understanding both deforestation processes and classifying modeling approaches. The five types of variables used in the 140 models of deforestation are: (1) The magnitude and location of deforestation; (2) the agents of deforestation, namely, individuals, households, or companies involved in land use change and their characteristics; (3) the choice variables (decisions about land allocation that determine the overall level of deforestation for the particular agent or group of agents); (4) Agents' decision parameters and (5) macroeconomic variables and policy instruments affecting forest clearing indirectly through their influence on the decision parameters Angelsen and Kaimowitz [56].

According to the authors, the agent or the source of deforestation (plantation companies, small farmers, etc.) has to be identified. Further, agents' decisions have to be considered, accounting for (1) their characteristics, including their preferences, seniority of a household head, gender and labor allocation as well as their initial resource, and (2) their decision parameters such as property regime, Agricultural commodity prices, timber prices, and income. These variables represent immediate or proximate causes. Finally, underlying variables, i.e., broader forces like macroeconomic variables or policy instruments that influence the source or agents and indirectly drive deforestation, have to be taken into account. Proximate drivers include human-induced factors that influence directly households' deforestation while underlying driving forces are fundamental social processes, that underpin the proximate causes and either operate at the local level or have an indirect impact from the national level [57, 58].

From their meta-analysis, Angelsen and Kaimowitz [56] derived two categories of models. Microeconomic models focus on immediate causes, and macroeconomic models deal with the underlying causes. They also suggest distinguishing between models based on perfect markets and models assuming imperfect markets.

Geist and Lambin [57] contributed to building this conceptual framework via a meta-analysis of 152 case studies culled from 95 publications. Their main contribution was the breakdown of numerous factors found in the existing literature into, (1) three aggregate proximate causes,

that is, agricultural expansion, wood extraction, and expansion of infrastructure; (2) five broad categories of underlying driving forces, that is, demographic, economic, technological, policy/institutional, cultural or socio-political factors; and a group of other variables associated with deforestation, comprising land characteristics, biophysical drivers and social trigger events (economic crises, war, etc.).

Combes et al. [59] contributed to the conceptual framework with a theoretical model that emphasizes a substitution effect between seigniorage and deforestation income. This contribution complies with the framework presented above. Indeed, Combes et al. [59] considered the triple Environment-Economic-Social crises, which Geist and Lambin [57] refer to as social trigger event, and proposed a link or a trade-off between macroeconomic and environmental outcomes, using an explicit model. This contribution is valuable to the traditional framework developed by Angelsen and Kaimowitz [56]. It presents a very feasible transmission channel between broad underlying drivers and deforestation. For instance, international transfers, public debt, and savings could be used by the government to optimize the inter-temporal allocation of natural resources and spending Combes et al. [59].

**2.1.2 Macro-level empirical studies.** There is a lot of information addressing the causes of tropical deforestation at national, regional, and global scales using macro-level data in developing countries, considering many types of forests, macroeconomic variables, institutional and policy factors [54, 55, 57, 59–69]. Major conclusions from a meta-analysis using results of 150 deforestation models by Kaimowitz and Angelsen [60] in Brazil, Cameroon, Costa Rica, Indonesia, Mexico, Thailand, Ecuador, the Philippines, and Tanzania indicate that deforestation tends to be greater when economic liberalization and adjustment policy reforms increase; when forested lands are more accessible; when agricultural and timber prices are higher; when rural wages are lower and there are more opportunities for long-distance trade. In Cameroon, Mertens et al. [68] and Sunderlin et al. [69] found that the annual rate of deforestation increased significantly in the decade after the economic crisis as compared to the previous period. They also found that the main proximate causes of deforestation were sudden rural population growth and the main underlying causes were macroeconomic shocks and structural adjustment policies. Nguyen Van and Azomahou [64] used a panel dataset of 59 developing countries over the 1972–1994 period to study the deforestation process. They found no evidence of an Environmental Kuznets Curve (EKC). They also pointed out political institution failures as factors that can worsen the deforestation process in developing countries. More generally, the evidence supporting the existence of an EKC for deforestation is contrasted [70].

Hosonuma et al. [30] derive deforestation and degradation drivers using empirical data synthesized from existing reports on national REDD+ readiness activities. They assessed the relative importance as well as the drivers of variability by continent between 2000 and 2010. They used the forest transition model, considering deforestation rate and remaining forest cover in 100 subtropical non-Annex I countries. They found that, similarly to Asia, the importance of deforestation drivers in Africa varies with different forest transition phases and with different areas. The impact of commercial agriculture on deforestation rises until the late-transition phase and the relative importance of subsistence agriculture remains fairly stable throughout the different phases.

**2.1.3 Micro-level empirical studies relating to agent livelihood decisions.** Over the past decade, there has been a rapid surge in the number of companies and multinational corporations making zero-deforestation commitments. This is publicly stated declaration of intent by private sector corporations to eliminate deforestation from their supply chains. There is strong evidence that forests and zero-deforestation commitments have an important role in ensuring livelihood and social outcomes over time and in some cases, contribute

to poverty alleviation [22, 71]. Some micro-level studies assess the degree of diversification/ specialization and related impact on poverty reduction. Others question the impacts of zero-deforestation commitments. Using the 2017 Ghana Living Standards Survey (GLSS7) from 14,009 households, Dagunga et al. [17] found that, while diversification lessens household poverty, the extent and dimension of diversification is important. Few studies have investigated the relation between agent livelihood decisions and tropical deforestation at the household level. Using the CIFOR-PEN dataset, comprising 7172 households from 24 developing countries, Babigumira et al. [72] analyze which household and contextual characteristics affect land use decisions in the developing world. The authors considered the sustainable livelihoods framework and assessed the role of various asset types on households' deforestation. The literature on the sustainable livelihoods framework (SLF) asserts that the ability to pursue different livelihood strategies depends on the basic material, social, tangible and intangible assets that people have in their possession (Scoones [16]). In different contexts, sustainable livelihoods can be achieved through access to natural, economic, human, physical, and social capital or resources. Babigumira et al. [72] found that 27% of rural households cleared forests for agricultural-based livelihoods. They also found that asset poverty does not drive deforestation. Indeed, households with medium to high asset holdings and higher market orientation were more likely to clear forests than the poorest and market-isolated households. Households that cleared forests were closer to the forest and came from villages with higher forest cover.

Relying on a rich panel dataset collected from the Tsimane communities in Bolivia, Perge and McKay [73] analyze the relationship between forest-based households' livelihood strategies, and forest clearing, and the relationship of both to welfare. Four livelihood strategies are identified, based on households' reported sources of cash earnings, namely, sale, wage, diversified and subsistence strategy. Forest clearing is positively linked to welfare, especially for households whose income results from combining agricultural sales and wage activities compared to households adopting other strategies. Households with a subsistence strategy are not able to accumulate assets in the long run. As one of the main conclusions, the authors state that households clear only small areas of forest with a positive effect on welfare, enabling the accumulation of assets.

Pacheco [74] define a typology of smallholders that accounts for livelihoods, farming systems, and wealth to analyze smallholders' deforestation in Uruará and Redenção in the Brazilian Amazon. The author uses household survey data from 136 interviews in Uruará and 82 interviews in the Redenção area and finds that cattle ranching is associated with a greater impact than cocoa or subsistence agriculture. Contrary to Perge and McKay [73], a strong correlation between deforestation and the wealth of the farmers is found.

**2.1.4 Spatial patterns studies.** Spatially explicit econometric studies of drivers of deforestation have taken more importance in the last few years [75]. These studies show that most deforestation tends to be located outside reserves and mountainous areas and deforestation occurs primarily within the more accessible Eastern counties and at areas near deforested areas.

Pfaff et al. [76] found evidence of spatial spillovers from roads in the Brazilian Amazon's deforestation. Considering local administrative entities, Amin et al. [77] found that deforestation activities of neighboring municipalities are correlated with some leakage. As a point of fact, protected areas may shift deforestation to neighboring municipalities.

Using a general spatial two-stage least squares model to analyze the determinants of deforestation in 24 Sub-Saharan African countries during the period spanning 1990 to 2004, Boubacar [78] found that deforestation in one country is positively correlated to deforestation in neighboring countries and that determinants of forest clearing are region specific.

## 2.2 Contribution

Data on tropical deforestation including household scale-level have frequently been questioned, considered unreliable or non-available [57, 61]; which makes reliable econometric studies on the drivers of deforestation difficult to implement in tropical Africa including Congo Basin countries [79]. While, unlike Southeast Asia and the Amazon regions, where large-scale agricultural operations play an important role, most deforestation in the Congo basin can be attributed to small-scale farmers using extensive slash-and-burn techniques [80, 81]. In the same vein, an original meta-analysis of 121 studies by Busch and Ferretti-Gallon [75] reveals a geographical lag of spatially explicit studies of tropical deforestation in Africa in peer-reviewed academic journals between 1996 and 2013, underlining the availability of data as the main constraint. In 2023, Busch and Ferretti-Gallon [82] updated their study with an additional review of 199 studies published between 2014 and 2019. They found an overall growth over time in spatially explicit econometric studies of deforestation, reforestation, and forest degradation, by region. Yet, in the three tropical forest basins, Africa experienced the lowest increase (12%) in the latter period, compared to Asia (31% increase), and Latin America (38% increase). Finally, an emerging study on Zero-Deforestation commitments (ZDCs) aims at understanding the effectiveness of these commitments in reducing deforestation and to characterize their potential impacts on rural livelihoods, on social sustainability criteria, on social outcomes, looking for possible strategies for achieving compliance with the social criteria [22]. There is a dearth of information on the relative contributions of specific activities like subsistence agriculture, internationally traded commodities, and diversified strategies, given the composition of activities' portfolios to deforestation.

In this context, our contributions are multiple:

- We assess the impact of livelihood choices on deforestation. To our knowledge, this research is among the pioneering studies that investigate the factors that govern households' deforestation in the Congo basin using a household-level survey. Indeed, west and central Africa account among regions that lag in econometric analysis of deforestation [57, 75]. At the same time, deforestation drivers are more complex and differ significantly across the world's regions, from one location or continent to another [19, 72, 83–85].

- We refer to the standard protocol of analyzing deforestation. This is crucial as it allows for improved comparisons in future research [57]. Our research considers and tests the influence of (1) agents' decision parameters such as family income and factor constraints (2) agents' characteristics and (3) other contextual variables such as choice and biophysical variables on the agent' deforestation. Further, our research used a microeconomic model, therefore, we focus on immediate causes as suggested by Angelsen and Kaimowitz [56]. Fig 1 shows the adaptation of our research to the conceptual framework for analyzing households' deforestation.

- We consider the interactions between people and wildlife and test the impact of human-wildlife conflicts on households' deforestation. We also test the impact of land conflict among households on households' deforestation.

- The influence of spatial spillovers is investigated. Besides direct effects on households' characteristics, we consider endogenous and exogenous interactions among households and test the possible resulting spillover effect on households' deforestation within their neighborhood.

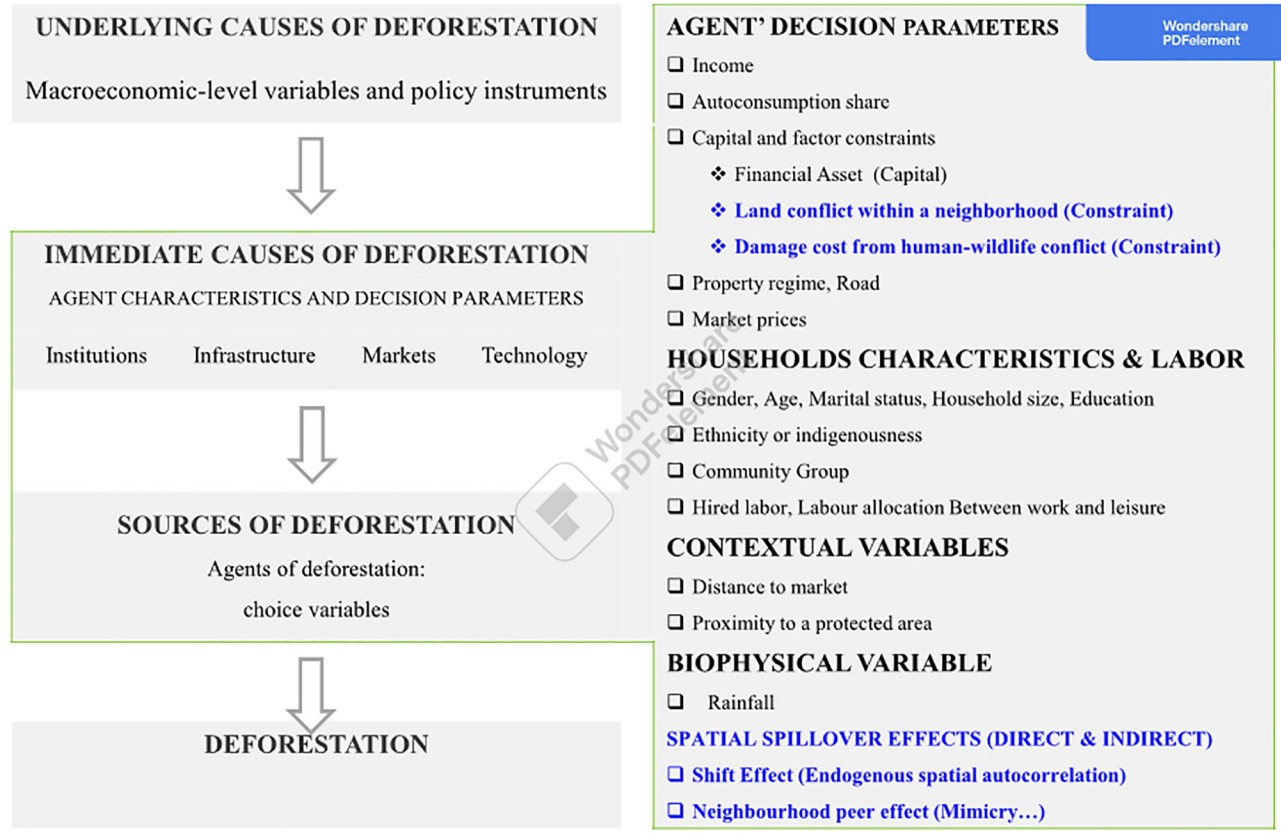

**Fig 1. Adaptation of deforestation framework to Tridom landscape case study. Source**: Authors, Adapted from Angelsen and Kaimowitz (1999). Elements in blue color represent our contribution to the framework.

## 3 Objective and hypothesis

This paper investigates the factors that drive households' deforestation in the Tridom landscape. It particularly considers the impact of households' choice of livelihood strategies, including cocoa production. It investigates the impact of human-animal conflict. It also considers spatial patterns as possible direct and indirect causes. More precisely, we test the following hypothesis:

### Household income given the livelihood strategies including cocoa production, influence deforestation

The impact of family income on small-scale deforestation is closely related to the households' livelihood strategies. This hypothesis will allow comparing the incremental change in households' deforestation resulting from a marginal increase in income given their livelihood portfolio and strategies. More precisely, we assume and test that internationally traded commodities, cocoa in this case, whether part of a specialized or a diversified strategy, drive larger deforestation compared to other livelihood strategies.

### Factor constraints such as both Human-Human and Human-Wildlife conflicts drive households' deforestation

As regard Human-wildlife interaction, about 259 households, that is 28% of the sample surveyed have experienced conflicts with elephants leading to CFA28,140 or $50 of damages cost

on average [86]. We hypothesize a higher likelihood for households experiencing Human-elephant conflict to look for additional or new land and thus, higher deforestation.

## Spatial patterns influence deforestation

This paper tests the presence of endogenous interaction of households' deforestation. Indeed, proximity among households in the Tridom landscape implies the existence of cultural and social interaction that could yield spatial spillover effects leading to similarities in deforestation decisions. Further, it was shown that households' deforestation as a social and cultural phenomenon is likely to be characterized by spatial autocorrelation [76, 78, 87–89]. The observations we did during the 8 months of fieldwork in Cameroon and Gabon reveal some competition about land holding among household heads. This observation calls for testing the existence of spatial effects within a household's neighborhood.

The paper considers a number of control variables. The first set of control variables includes distance to market, distance to protected areas, financial asset, and households' characteristics. We refer to financial assets as Cartas & Harutyunyan [90]. Financial assets are financial instruments or financial claims arising from contractual relationships with the basis of creditor/debtor relationships. We then consider Loans, and money transfers/remittances.

Following Caldas et al. [91], Fontes and Palmer [92], and Pfaff [54] distance to market influences deforestation. Distance to market is considered here as an indicator of transaction cost regarding land location. Environmental state and policy, captured by the distance to the nearest protected area, rank among households' deforestation drivers. Due to their legal status, protected areas are supposed to be associated with lower deforestation [75, 93]. Yet they may also have an impact on surrounding deforestation, for instance through leakage [77]. Financial asset drives households' deforestation [94]. Households' characteristics influence deforestation. Gender, household head age and education, marital status, household size, ethnicity as well as the duration of residence (seniority) account among the drivers of small-scale deforestation.

The second group of control variables includes Households' Choice variables such as labor, social assets, and biophysical variables. Following Pfaff et al. [95]; Walker et al. [94], labor allocation between work and leisure and hired labor increases forest clearing. In our study area, hired labor is most often made up of Baka indigenous people. They are employed at a very low cost. We also test that social capital, like belonging to a group of interest, has an impact on households' deforestation. (3) Finally, following Chowdhury [96, 97] we test that biophysical variables, namely rainfall, have a strong impact on small-scale deforestation.

## 4 A simple microeconomic model of deforestation choices

Consider household $i$ choosing his/her level of deforestation $D_i$ to maximize its utility:

$$\max_{D_i} U_i(D_i, L_i, X_i, D_j) \tag{1}$$

$L_i$ is the livelihood strategy selected by the household, as defined in Ngouhouo-Poufoun et al. [98]. Six different household strategies are considered: Subsistence agriculture ($A$), cocoa crops ($C$), forest-based activities ($F$), and combinations of cocoa/forest-based ($CF$), agriculture/forest-based ($AF$) and agriculture/cocoa/forest-based ($ACF$); such that $L_i$ = [$A$, $C$, $F$, $CF$, $AF$, $ACF$].

$X_i$ is a vector of household $i$ socio-economic control variables susceptible to influence deforestation. Household $i$'s utility function may encompass income, but also other non-observable outcomes such as household vulnerability. Thus, the household characteristics $X_i$ may

influence not only the household economic return but also other household matters of interest. Furthermore, we also consider that household $i$'s utility may be influenced by its neighbors. $D_j$ is the level of deforestation chosen by household $i$'s neighbors, which is likely to influence its decision. This type of strategic interaction is close to the resource-flow model presented by Brueckner [88] and Anselin [89].

The first-order condition implicitly gives the optimal level of deforestation $D_i^*(L_i, X_i, D_j^*)$ for household $i$:

$$U'_{D_i} = \frac{\partial U_i(D_i^*, L_i, X_i, D_j^*)}{\partial D_i^*} = 0 \qquad (2)$$

Optimal deforestation strongly depends on livelihood strategies chosen by the households:

$$D_i^*(L_i, X_i, D_j^*) \neq D_i^*(L_i', X_i, D_j^*), \quad \forall L_i \neq L_i'. \qquad (3)$$

Moreover, one can then infer the impact of livelihood strategies, other variables, and neighbors deforestation on household $i$ deforestation level:

$$\frac{\partial D_i^*(L_i, X_i, D_j^*)}{\partial X_i} = -\frac{\frac{\partial U'_{D_i}}{\partial X_i}}{\frac{\partial U'_{D_i}}{\partial D_i}} \qquad (4)$$

$$\frac{\partial D_i^*(L_i, X_i, D_j^*)}{\partial D_j} = -\frac{\frac{\partial U'_{D_i}}{\partial D_j}}{\frac{\partial U'_{D_i}}{\partial D_i}} \qquad (5)$$

In the next section, we will investigate the impact of livelihood choices on deforestation levels (sign of Eq (3)), the impact of other control variables (sign of Eq (4)) and the nature of spatial spillovers (sign of Eq (5)).

## 5 Spatial econometric procedure and data

The common observation that individuals belonging to the same group tend to behave similarly can be explained by three hypotheses of the standard linear model (SLM) that are the endogenous effects, the exogenous effects, and the correlated effects [99]. The endogenous and exogenous effects express distinct ways that persons might be influenced by their social environments. The first assumes that all else equal, individual behavior (deforestation ($D_i$) tends to vary with the average behavior (deforestation of the group or neighbor ($D_{-i}$)). The second effect assumes that individual behavior is in some way influenced by the characteristics of the group or neighbors ($Z_{-i}$). The correlated effects express non-social phenomena. Similarities in individuals' behavior may results from spatially dependent omitted variables, interaction among error terms ($\epsilon$) or environmental similarities [88, 99–101]. In the following, we present a short description of various cross-sectional spatial econometric models (5.1). We present the selection procedure we used (5.2). Then, we present the data used in the econometric procedure (5.3).

## 5.1 Cross-sectional spatial econometric models

The matrix form of the generalized nested spatial model that accounts for all three effects was defined by Manski [99] in Eqs (6) and (7). This model is also called the Manski model.

$$D = \alpha * I_N + \rho WD + Z\beta + WZ\theta + \epsilon, \tag{6}$$

$$\epsilon = \lambda W\epsilon + \mu \tag{7}$$

In this expression, $I_N$ is a n by n identity matrix. $WD$ denotes the endogenous effects, representing the average deforestation of neighboring individuals ($D_{-i}$). The $\rho$ parameter measures the strength of spatial dependence. $W$ is a row-standardized weights matrix such that the elements ($w_{ij}$) in each row ($i$) sum to one and the diagonal elements set to zero, each element ($w_{ij}$) measures the intensity of interaction among household's ($i$) and its relevant neighbors [102]. $WZ$ stands for the exogenous effects representing the average value of neighboring households' characteristics, scaled by the parameter $\theta$. The parameter $\beta$ captures the direct impact of independent variables. $W\epsilon$ denotes the interaction among the disturbance terms. $\lambda$ measures the spatial autocorrelation intensity among error terms. After testing the Eqs (6) and (7), Manski [99] found that data on equilibrium outcomes cannot distinguish both endogenous and exogenous interactions from contextual effects based on testing the model (6) and (7).

Further, Lesage [103] suggested specifying a model that accounts for both endogenous and exogenous spatial effects among individuals. Eq (8) is the resulting model, called the Spatial Durbin Model (SDM) by Anselin [101]. This model is equivalent to the component (6) of the Manski model, with $\lambda = 0$ in (7). Following [104], The SDM will allow the deforestation of each household to vary with respect to both its own characteristics and the mean characteristics within his/her neighborhood.

$$D = \alpha * I_N + \rho WD + Z\beta + WZ\theta + \epsilon \tag{8}$$

A year later, Kelejian and Prucha [105] suggested including both endogenous interaction effects and correlated effects among the error terms. This model is equivalent to the Manski equation with $\theta = 0$ in the component (6). This model is called the Kelejian-Prucha Model or the Spatial Autoregressive model with Autoregressive disturbances (SARAR). This allows spatial autocorrelation in both non-observed patterns and households' deforestation, without spillover effects neither from the neighborhood characteristics nor from own characteristics on neighboring households.

$$D = \alpha * I_N + \rho WD + Z\beta + \epsilon \tag{9}$$

$$\epsilon = \lambda W\epsilon + \mu. \tag{10}$$

The Spatial Autoregressive Model (SAR) was proposed by Anselin [101] to test only the endogenous interaction, using the lag value of the dependent variable. The SAR model allows only spatial autocorrelation of households' deforestation, without spillover effects neither from the neighborhood nor from own behavior on neighboring households.

$$D = \alpha * I_N + \rho WD + Z\beta + \epsilon \tag{11}$$

Among other models, (1) the Spatial Error Model (SEM) was developed by Anselin [101] to account only for the correlated effects. This assumes that $\rho = 0$ in the SDM model. (2) The Spatial Durbin Error Model (SDEM) includes spatial lags of independent variables ($\theta \neq 0$) and the

spatially lagged error term ($\lambda \neq 0$) in Eqs (6) and (7). (3) The Spatial Lag of the explanatory variable (SLX), ($\rho = 0$), ($\theta \neq 0$) and ($\lambda = 0$) in Eqs (6) and (7).

## 5.2 Selection procedure

The consideration of spatial effects in econometric models requires some specific processes to avoid model misspecification [106]. The usual standard approach is to start with a specific SLM. Further, test if the error terms and/or the dependent variable are spatially correlated, to specify the spatial model that is consistent with the data generation process. This is called the specific-to-general approach. The second approach is to start with the Mansky model and test progressively the existence of various spatial effects [100]. In this study, we started with the standard approach that is most common in spatial analyses, following Anselin [107]. After estimating an SLM, we first tested for the existence of spatial autocorrelation using the Moran *i* statistic on the residual of the linear model. Further, we proceeded to the Lagrange Multiplier test which helps to find the type of spatial effects that fit with our data generation process. Tablos 2 and 3 in the subsection 6.1 display our procedure of model specification.

An issue that arises in applied econometrics is the need to compare models [104]. Indeed, a universal criticism of spatial regression models is the sensitivity of the estimates and inferences to the form of spatial weight matrix [108]. After specifying our econometric model, we use four types of weight matrix namely, the Gabriel graph weight matrix, the five nearest neighbors (5NN), the ten nearest neighbors (10NN), and the distance-based weight matrix to account for this criticism. These weight matrices are presented in detail in Ngouhouo-Poufoun et al. [98].

## 5.3 Data, variables and descriptive statistics

**5.3.1 Data.** We carried out a face-to-face stratified survey of 1035 households out of about 64,140 within the Tridom landscape. The respondents were selected randomly to ensure statistical representativeness. The Tridom landscape is the Dja-Minkebe-Odzala Tri-national forest conservation landscape spanning Cameroon, Gabon, and the Republic of Congo. It covers a geographical area of 191,541 km2. That is about 7.5% of the Congo Basin forest located in Central Africa. For further information on the study area, the choice of respondents, the sample distribution, and the survey administration see Ngouhouo-Poufoun et al. [86].

**5.3.2 Dependent variables.** In this study, deforestation stands for forest conversion to any agricultural activities including food and perennial crop systems, as well as fallow, during the past decade. Our measure of deforestation followed two steps. We first asked the households to fill in the information about their total land-holding during the past decade in a table. After that, we randomly chose one plot among the total plots declared by the household to visit. The visited plot was tracked using a Global Positioning System (GPS) to have the real area. A total of 3338 plots of land were declared by the overall sample which is on average 3.2 plots held by each household. A total of 526 plots were tracked with a GPS. To avoid the unreliability of recall data, the data declared were adjusted using the tracked data to obtain the value used in this study. On average, household heads declared have cleared 4.75 ha of forest for agricultural land use, either small-scale subsistence farming or cash-crop such as cocoa. We found after statistical adjustment that the average land clearing of each household stands at 4.41 ha. Fig 2 displays some indicators of deforestation relating to livelihood strategies.

**5.3.3 Independent variables.** We distinguish our independent variables into three categories. First, we consider the household income, depending on the livelihood strategy chosen. In

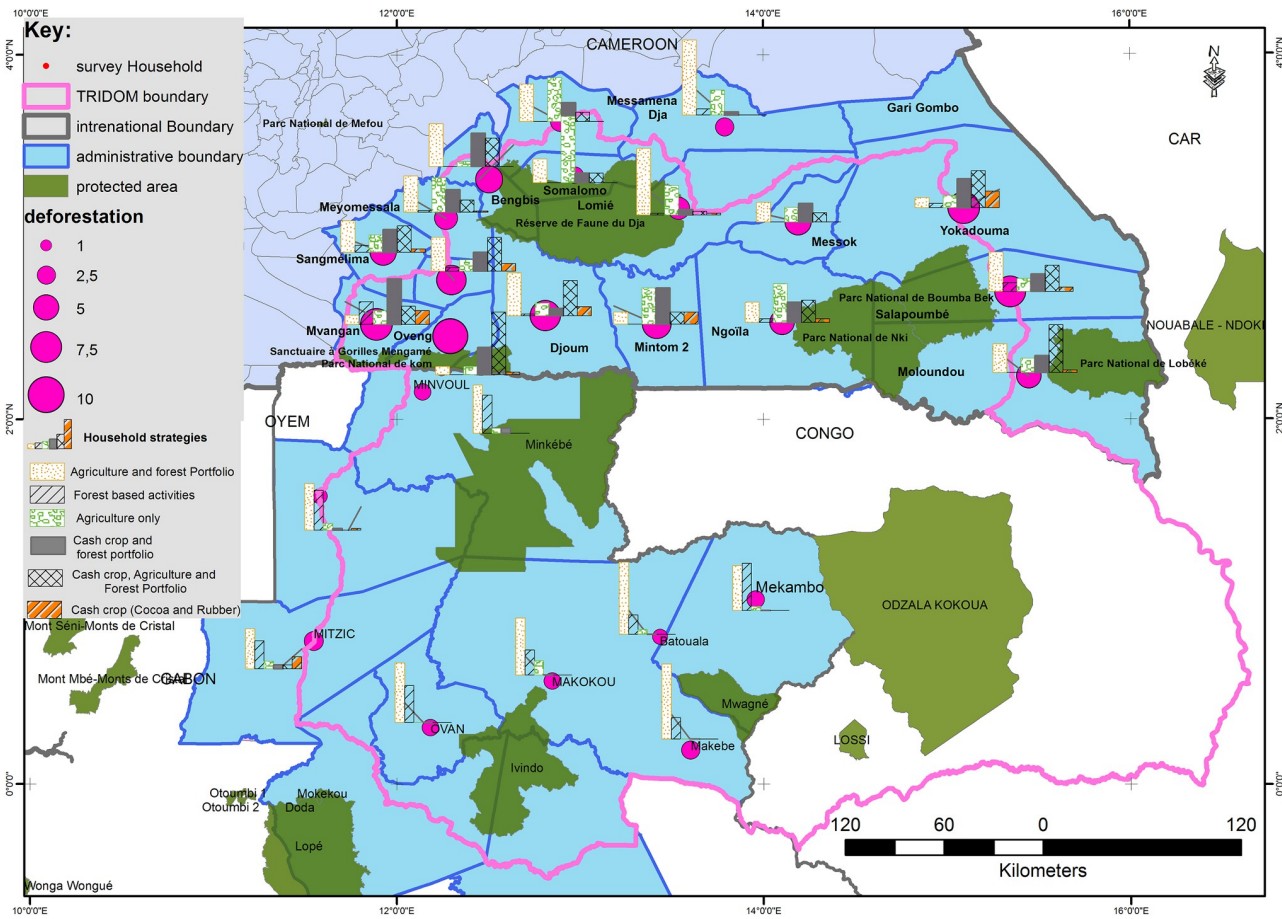

**Fig 2. Livelihoods and mean deforestation in the landscape. Source of data: Protected area, TRIDOM limit, Central Africa, Administrative limit**:
Country-specific land-use and administrative data come from the Forest Atlases published by the forest ministries (Ministry of Forests and Fauna in
Cameroon, Ministry of Water, forests, the Sea and the Environment, in charge of the climate plan and the land use plan in Gabon, and Ministry of
Forest Economy in the Republic of Congo). All these are Open Data, and were collected, digitalized, harmonized, and published by World Resources
Institute (www.wri.org) in 2014, and are also available on the Forest Atlas platform and Global Forest Watch. Cameroon Forest Atlas, the Ministry of
Forest and Fauna/World Resources Institute accessed on (10/03/2023), https://cmr.forest-atlas.org/. Republic of Congo Forest Atlas, Ministry of Forest
Economy in the Republic of Congo /World Resources Institute was accessed on (10/03/2023), https://cog.forest-atlas.org/. Gabon Forest Atlas, Ministry
of Water, forests, the Sea, and the Environment, in charge of the climate plan and the land use plan /World Resources Institute, accessed on (10/03/
2023), https://gab.forest-atlas.org/pages/maps **Deforestation:** Data collected in the field in Gabon and Cameroun during the fieldwork carried out by
the Authors in 2013—2014. A total of 526 plots were tracked with a Global Positioning System (GPS) to have the real area. **Household strategies:**The
income from livelihoods strategies in the map are calculated by the authors using data collected during the fieldwork carried out by the Authors in 2013
—2014 in Gabon and Cameroun.

our case study, like many contexts in rural areas of developing countries, access to land can be
considered open.

Second, several potential constraints to deforestation are considered: Credit constraints are
approximated through credit and money transfer received by the household; land use conflict
is a proxy for constraints on land access; while the damage costs from wildlife conflicts repre-
sent environmental damages; proximity to protected areas represents constraints brought by
environmental policies.

Third, diverse households' characteristics and contextual variables are used, such as dis-
tance to markets, as a proxy for transaction costs. Table 1 displays the variables' definitions
and descriptive statistics.

**Table 1. Variables and descriptive statistics (N = 986 households).**

| Variable | Definition of variables | Mean | Std. Dev. |
|---|---|---|---|
| **AGENT'S DECISION PARAMETERS** | | | |
| **Income from diversification livelihood strategies (FCFA10$^3$)** | | | |
| *ACF_households* | Income from mixing Agriculture & Cocoa & Forest | 38.96 | 117.06 |
| *AF_households* | Income from mixing Agriculture & Forest | 54.97 | 126.6 |
| *CF_CocoaForest* | Income from mixing Cocoa & Forest | 21.28 | 73.34 |
| **Income from specialization livelihood strategies (FCFA10$^3$)** | | | |
| *F_Forestbased* | Income from specializing in Forest-based activities | 70.18 | 372.79 |
| *A_Agriculture* | Income from specializing in Agricultural | 12.39 | 121.41 |
| *C_Cocoa* | Income from specializing in Cocoa | 8.85 | 67.93 |
| **Other decision variables** | | | |
| Autocons_Share (% of total value) | Autoconsumption share in the total income | 0.26 | 0.19 |
| **Capital & factor constraints** | | | |
| Finance_asset | Credit and money transfer (CFAF/month) | 8.67 | 33.55 |
| Human_Wildlife (FCFA10$^3$) | Damage cost of wildlife conflict (CFA/month) | 0.62 | 1.45 |
| Landconflict Dummy (1 = yes) | Land use conflict, Dummy (1 = yes) | 0.18 | 0.38 |
| **HOUSEHOLDS CHARACTERISTICS** | | | |
| Gender | Gender, Dummy (1 = Male) | 0.77 | 0.42 |
| Age | Household head age (continuous, in years) | 48.44 | 14.61 |
| Ages_thr | Age centered and squared | 213.34 | 246.72 |
| Marit_single | Matrimonial status, Dummy (1 = Maried) | 0.7 | 0.46 |
| Hsize | Household size (continuous) | 6.5 | 4.01 |
| Schoolcycl_2 | Education level, Dummy (1 = secondary school) | 0.56 | 0.5 |
| Autochbaka | Indegenouesness, Dummy (1 = Baka. 0 = Bantou) | 0.05 | 0.22 |
| Seniority | Seniority in the village (continuous, in years) | 27.01 | 20.71 |
| CommunityGroup | Community Interest Companies, Dummy (1 = yes) | 0.28 | 0.45 |
| Baka_employmt | Baka employment (continuous) | 1.87 | 2.96 |
| laborduration | Working hour per day | 5.49 | 4.39 |
| **CONTEXTUAL VARIABLES** | | | |
| Country | Country, Dummy (1 = Cameroon, 0 = Gabon) | 0.73 | 0.44 |
| Distmarket | Distance to market (in Km) | 65.06 | 58.69 |
| Distance to P. Areas | Distance to the nearest Protected Area (in Km) | 29.3 | 22.58 |
| **Biophysical factor** | | | |
| Rainfall | The per annum amount of rain that falls (mm) | 1638.30 | 113.66 |

# 6 Results

## 6.1 Spatial dependence and endogeneity diagnosis

Table 2 below displays the Moran coefficient index computed after running the SLM. This statistic tests the existence of spatial autocorrelation. Except for the Gabriel Graph weight structure, the index value is positive and statistically greater than 0. It appears a positive spatial clustering of deforestation among nearby households in the Tridom landscape.

The Lagrange multiplier test presented in Table 3 is used to diagnose the type of spatial dependence that governs our data generation process among the endogenous effects, i.e. spatial lag of the dependent variable ($\rho \neq 0$) and the correlated effects or the spatial autocorrelation of the disturbance term ($\lambda \neq 0$). This test suggests rejecting all the specifications that allow spatial autocorrelation in the disturbance term. Therefore, we avoid estimating the SARAR,

**Table 2. Spatial autocorrelation test.**

| | Global Moran I | | | | Moran I test under randomization | | | |
|---|---|---|---|---|---|---|---|---|
| | Moran I | E(I) | z(I) | P-value | Moran I | E(I) | z(I) | P-value |
| gabhsld.w | 0.0171 | -0.0032 | 0.7060 | **0.2401** | 0.1191 | -0.0010 | 4.2133 | **0.0000** |
| 3NN weight matrix | 0.0377 | -0.0032 | 1.7138 | **0.0433** | 0.1461 | -0.0010 | 6.2109 | **0.0000** |
| 4NN weight matrix | 0.0339 | -0.0032 | 1.7903 | **0.0367** | 0.1396 | -0.0010 | 6.8335 | **0.0000** |
| 5NN weight matrix | 0.0336 | -0.0032 | 1.9876 | **0.0234** | 0.1301 | -0.0010 | 7.1017 | **0.0000** |
| 10NN weight matrix | 0.0363 | -0.0030 | 2.9635 | **0.0015** | 0.1352 | -0.0010 | 10.2690 | **0.0000** |
| 17NN weight matrix | 0.0217 | -0.0028 | 2.4270 | **0.0076** | 0.1218 | -0.0010 | 12.0470 | **0.0000** |
| Distance based weight matrix | 0.0171 | -0.0025 | 3.1368 | **0.0009** | 0.0993 | -0.0010 | 15.4710 | **0.0000** |

the SEM, and the generalized nested Manski spatial model. In the following, we estimate the SAR as it fits with our data generation process. Comparing the Akaike information criterion (AIC) and Bayes' information criterion (BIC), we confirm the SAR model rather than the SDM and SEM models, cf. S3 Table. Indeed, the test displays lower BIC and AIC estimates for SAR compare to SEM and SDM. The relative amount of information lost by the SAR model is then lower than the amount lost by SEM and SDM models. We then prefer the SAR model with lower AIC and BIC. We avoid displaying the results of the SDM as it yielded counter-intuitive findings.

Overall, following the findings in Tables 2 and 3, we cannot reject our hypothesis of spatial effects, giving rise to a presumption of the existence of a positive relation between households' deforestation and the average deforestation of neighboring households. Section 6.2 below will confirm or reject this presumption via the $\rho$ parameter and present the drivers of households' deforestation.

Our analysis might be subject to endogeneity bias resulting from a simultaneous relation between the dependent variable and some independent variables, resulting in inconclusive and inconsistent findings [109–112].

Additional to the spatial lag deforestation which is endogenous to the dependent variable since it implies simultaneous spatial interaction ($D = f(WD)$), simultaneity between deforestation and some of our control variables might also arise, in particular those related to income from all the livelihood strategies comprising cocoa, whether specialized or diversified.

Indeed, among the livelihood strategies, cocoa production requires higher area of land compared to other strategies, and as a cash-crop, it is supposed to generate higher income. Yet, higher income from cocoa might lead to higher demand of land to expand cocoa production.

**Table 3. Lagrange multiplier diagnostic of spatial dependence.**

| | LM Test for Spatial Error Components | | | | LM Test for Spatial lag model | | | | |
|---|---|---|---|---|---|---|---|---|---|
| | Ordinary LMerr | | Robust LMerr | | Ordinary LMlag | | Robust LMlag | | |
| | | | ($\rho = 0$) | | | | ($\lambda = 0$) | | |
| | Stat. | P-value | Stat. | P-value | Stat. | P-value | Stat. | P-value | |
| Gabhsldweight matrix | 0.3512 | 0.5535 | 0.1233 | **0.7254** | 0.6565 | 0.4178 | 0.4286 | **0.5127** | |
| 3NN weight matrix | 2.4699 | 0.1160 | 0.5971 | **0.4397** | 4.4385 | 0.0351 | 2.5657 | **0.0992** | * |
| 4NN weight matrix | 2.6354 | 0.1045 | 0.6091 | **0.4351** | 4.9138 | 0.0266 | 2.8875 | **0.0893** | * |
| 5NN weight matrix | 3.2194 | 0.0728 | 0.1058 | **0.7449** | 4.9313 | 0.0264 | 1.8178 | **0.0976** | * |
| 10NN weight matrix | 7.2122 | 0.0072 | 0.8022 | **0.3704** | 13.0090 | 0.0003 | 6.5990 | **0.0102** | ** |
| 17NN weight matrix | 4.3451 | 0.0371 | 2.0311 | **0.1541** | 12.2670 | 0.0005 | 9.9533 | **0.0016** | *** |
| Distance based weight matrix | 6.4858 | 0.0109 | 0.0442 | **0.8335** | 12.0590 | 0.0005 | 5.6177 | **0.0178** | ** |

Such a mechanism is all the more true that the cocoa strategy of the Cameroonian government is to raise the national production from 292.471 million kg during the 2020–2021 campaign to 640,000 tonnes by 2030. One may therefore suspect a potential causal relationship between strategies based on cocoa and relating income and deforestation. Higher level of income from cocoa activities may imply higher land holding and *vice versa*, hence the suspicion of the endogeneity bias related to the income resulting from cocoa variables.

Despite the above, cocoa yield is very low in our study area (see S2 Fig). A higher level of land acquisition does not necessarily imply a higher level of income. One may be likely to exclude possible causal relations as well as relating endogeneity bias. In addition, only a few studies take into account the simultaneous presence of spatial autocorrelation and endogenous explanatory variables (cf. [113] for more details). Yet, to completely clear the risk of endogeneity or simultaneity bias with one or more independent variables, we carried out a robustness analysis using an instrumental variable method to test empirically this simultaneity bias [111] (cf. S2 Table). The Wu-Hausman test suggests not confirming the endogeneity of income cocoa variables. Furthermore, the null hypothesis of weak instruments is strongly rejected implying that our instruments are robust. Sargan's test of the over-identification of restrictions on instruments was not significant, which implied that our instruments are valid. As in S3 Fig, a correlogram was used to control for multicollinearity.

## 6.2 What are the immediate causes of households' deforestation in the Tridom landscape?

**6.2.1 Robustness check and spatial dependence.** Table 4 displays the estimated results based on the SLM and four variants of the SAR model considering four different types of the weight matrix. This result considers only the factors that significantly drive households' deforestation. S1 Table displays the results from the model with overall variables presented in the subsection 5.3 above. Insignificant variables were removed progressively until we got the reduced set of significant variables. The post-estimation tests (Akaike Information Criterion (AIC), Wald, and LR tests) confirm the reduced model as best-suited compared to the full model displaid in S1 Table.

The SAR models show a significant spatial dependence between the deforestation of each household and the average deforestation of neighboring households. This suggests some similarities in deforestation decisions of households located nearby. The expected deforestation of each household in the Tridom landscape is determined by both its own characteristics and a linear combination of neighboring households' deforestation scaled by $\rho$. The SLM estimates have a larger size compared to the SAR models considering all types of weight matrices. It attributes the variability in households' deforestation only to the independent variables. Also, the SAR model suggests that the variability of deforestation across households is partially explained by neighbors' deforestation behavior. Further, the spatial lag of households' deforestation is treated as an endogenous variable and the error term is influenced by the same process. As a result, although the Q-Q plot in S1 Fig reveals the normal distribution of households' deforestation, the SLM is biased and yields inconsistent estimates due to simultaneity bias. In these conditions, the SAR is a proper specification to account for this endogeneity [114]. Following Anselin [101, 114] our spatial lag model of deforestation was estimated using the maximum likelihood technique.

As shown in Table 4, the strength of spatial dependence ($\rho$) varies along with the type of the weight matrix. The scale of the $\rho$ parameter varies increasingly from 0.027 for the Gabriel graph weight matrix to 0.235 for the distance-based weight matrix. It equals 0.089 for the 5NN and 0.179 for the 10NN weight matrices. Further, the estimates vary decreasingly from the

**Table 4. Spatial autoregressive model.**

| | SLM | | | GabGraph | | | 5NN Weight matrix | | | 10NN Weight Matrix | | | Distance based | | |
|---|---|---|---|---|---|---|---|---|---|---|---|---|---|---|---|
| | Coef. | SD | | Coef. | SD | | Coef. | SD | | Coef. | SD | | Coef. | SD | |
| (Intercept) | -3.5941 | 1.1117 | *** | -1.1331 | 0.7166 | | -1.2545 | 0.7185 | * | **-1.4842** | **0.7173** | ** | -1.7664 | 0.7390 | ** |
| **AGENT'S DECISION PARAMETERS** | | | | | | | | | | | | | | | |
| **Income from diversification livelihood strategies (FCFA10³)** | | | | | | | | | | | | | | | |
| ACF_households | 0.0109 | 0.0013 | *** | 0.0108 | 0.0013 | *** | 0.0106 | 0.0013 | *** | **0.0102** | **0.0013** | *** | 0.0102 | 0.0013 | *** |
| AF_households | 0.0026 | 0.0012 | ** | 0.0026 | 0.0012 | ** | 0.0025 | 0.0011 | ** | **0.0025** | **0.0011** | ** | 0.0025 | 0.0011 | ** |
| CF_CocoaForest | 0.0120 | 0.0021 | *** | 0.0119 | 0.0020 | *** | 0.0116 | 0.0020 | *** | **0.0112** | **0.0020** | *** | 0.0113 | 0.0020 | *** |
| **Income from specialization livelihood strategies (FCFA10³)** | | | | | | | | | | | | | | | |
| A_Agriculture | 0.0029 | 0.0012 | ** | 0.0029 | 0.0012 | ** | 0.0029 | 0.0012 | ** | **0.0029** | **0.0012** | ** | 0.0028 | 0.0012 | ** |
| C_Cocoa | 0.0186 | 0.0021 | *** | 0.0185 | 0.0021 | *** | 0.0183 | 0.0021 | *** | **0.0180** | **0.0021** | *** | 0.0179 | 0.0021 | *** |
| Autocons_Share (% of total value) | -2.1269 | 0.7737 | *** | -2.1210 | 0.7667 | *** | -2.1518 | 0.7649 | *** | **-2.0922** | **0.7620** | *** | -2.1346 | 0.7633 | *** |
| **Capital & factor constraints** | | | | | | | | | | | | | | | |
| Finance_asset (FCFA10³) | 0.0106 | 0.0043 | ** | 0.0106 | 0.0042 | ** | 0.0104 | 0.0042 | ** | **0.0102** | **0.0042** | ** | 0.0099 | 0.0042 | ** |
| Human_Wildlife (FCFA10³) | -0.2135 | 0.0979 | ** | -0.2125 | 0.0970 | ** | -0.2143 | 0.0968 | ** | **-0.2179** | **0.0965** | ** | -0.2183 | 0.0966 | ** |
| **HOUSEHOLDS CHARACTERISTICS** | | | | | | | | | | | | | | | |
| Gender (1 = Male) | 0.6200 | 0.3450 | * | 0.6127 | 0.3419 | * | 0.5928 | 0.3411 | * | **0.5816** | **0.3398** | * | 0.6210 | 0.3404 | * |
| Age (continuous. in years) | 0.0293 | 0.0115 | ** | 0.0293 | 0.0114 | ** | 0.0291 | 0.0114 | ** | **0.0301** | **0.0113** | *** | 0.0303 | 0.0114 | *** |
| Ages_thr | -0.0013 | 0.0006 | ** | -0.0013 | 0.0006 | ** | -0.0013 | 0.0006 | ** | **-0.0013** | **0.0006** | ** | -0.0013 | 0.0006 | ** |
| Hsize (continuous) | 0.1692 | 0.0379 | *** | 0.1677 | 0.0376 | *** | 0.1669 | 0.0375 | *** | **0.1650** | **0.0374** | *** | 0.1675 | 0.0374 | *** |
| Seniority (continuous. in years) | 0.0401 | 0.0080 | *** | 0.0402 | 0.0079 | *** | 0.0404 | 0.0079 | *** | **0.0399** | **0.0078** | *** | 0.0390 | 0.0079 | *** |
| CommunityGroup Dummy (1 = yes) | 0.5566 | 0.3263 | * | 0.5605 | 0.3233 | * | 0.5777 | 0.3226 | * | **0.5718** | **0.3214** | * | 0.5922 | 0.3220 | * |
| Baka_employmt (coutinuous) | 0.1428 | 0.0507 | *** | 0.1462 | 0.0503 | ** | 0.1522 | 0.0502 | *** | **0.1584** | **0.0500** | *** | 0.1539 | 0.0501 | *** |
| **CONTEXTUAL VARIABLE** | | | | | | | | | | | | | | | |
| Country (1 = Cameroun. 0 = Gabon) | 1.3424 | 0.3556 | *** | 1.2738 | 0.3617 | *** | 1.0840 | 0.3722 | *** | **0.8565** | **0.3826** | ** | 0.8475 | 0.4080 | ** |
| R-squared: | 0.32 | | | | | | | | | | | | | | |
| F-statistic: (16; 969) | 28.38 | | * | | | | | | | | | | | | |
| **Rho ($\rho$)** | | | | 0.0274 | | | 0.0892 | | ** | **0.1799** | | *** | 0.2354 | | *** |
| Log Likelihood | | | | -2856 | | | -2855 | | | **-2852** | | | -2853 | | |
| ML residual $\sigma$ | | | | 4.39 | | | 4.38 | | | 4.36 | | | 4.3667 | | |
| AIC Criterion | 5750 | | | 5751 | | | 5748 | | | **5742** | | | 5744 | | |
| Wald Statistic | | | | 0.737 | | | 3.996 | | ** | **10.399** | | *** | 7.848 | | *** |
| LR test value | | | | 0.694 | | | 4.207 | | | **10.165** | | | 8.010 | | |
| LM for Residual autocorrelation | | | | 0.122 | | | 0.262 | | | **1.014** | | | 0.128 | | |
| Observations | 986 | | | 986 | | | 986 | | | 986 | | | 986 | | |

*, ** and *** = significance level at 1% 5% 10% respectively

Gabriel graph matrix to the distance-based matrix. The warning in Lesage [108] regarding the sensitivity of the estimates and inferences to the type of matrix is confirmed. Among these four candidate models, the 10NN base model, displayed in the third column with bold characters, performs better as it minimizes information loss. This model has the maximum log-likelihood with the minimum Akaike Information Criterion (AIC) compared to others. The goodness-of-fit test confirms that the SAR model based on the 10NN weight matrix is the best to fit the households' deforestation. Indeed, combining the Wald test (*W*), the Log-likelihood Ratio test (*LR*) and the Lagrange Multiplier test (*LM*) as suggested by Anselin [101], we found that the inequality $W \geq LR \geq LM$ is verified only for the 10NN based model that is

$(W = 10.399) \geq (LR = 10.16) \geq (LM = 1.014)$. In the following, estimates from the SAR model based on the 10NN weight matrix are used to derive the drivers of households' deforestation.

The $\beta$ coefficient of the SAR model cannot be interpreted as partial derivatives of households' deforestation in the Tridom landscape with respect to a one-unit change of various independent variables as in conventional linear regression model [115]. The subsection 6.2.2 below presents the impact of the various independent variables on the households' deforestation.

**6.2.2 Direct, indirect and total effects.**  Table 5 displays the factors that proximately drive households' deforestation in the Tridom landscape. These factors are regrouped into (1) Livelihood Strategies; (2) household characteristics and (3) contextual variables. Variables with insignificant coefficients are displayed in the full model in S1 Table.

**livelihood strategies**: The direct effects of household income are all positive and significant regardless of the livelihood strategy, except for the income of households who practice forest-based activities.

Diversification strategies and specialization strategies have comparable impacts on deforestation. Strategies that encompass cocoa production have the highest impact. On the other hand, the family income of households specializing in forest-based activities unsurprisingly does not tend to impact deforestation. Further, the indirect effects of these incomes are positive and significant except for households who adopt a diversified portfolio comprised of agriculture and forest-based activities. More precisely:

> **Table 5. Direct, indirect and total effects.**

| | 2*Coeff. | Direct Effects | | | Indirect effects | | | Total effects | | |
|---|---|---|---|---|---|---|---|---|---|---|
| | | Mean | SD | | Mean | SD | | Mean | SD | |
| **AGENT'S DECISION PARAMETERS** | | | | | | | | | | |
| **Income from diversification livelihood strategies (FCFA10³)** | | | | | | | | | | |
| ACF_households | 0.0102 | 0.0102 | 0.0012 | *** | 0.0023 | 0.0009 | *** | 0.0125 | 0.0017 | *** |
| AF_households | 0.0025 | 0.0025 | 0.0012 | ** | 0.0006 | 0.0003 | | 0.0030 | 0.0014 | ** |
| CF_CocoaForest | 0.0112 | 0.0113 | 0.0020 | *** | 0.0025 | 0.0010 | ** | 0.0138 | 0.0025 | *** |
| **Income from specialization livelihood strategies (FCFA10³)** | | | | | | | | | | |
| A_Agriculture | 0.0029 | 0.0030 | 0.0012 | ** | 0.0007 | 0.0004 | * | 0.0036 | 0.0015 | ** |
| C_Cocoa | 0.0180 | 0.0180 | 0.0021 | *** | 0.0040 | 0.0016 | *** | 0.0220 | 0.0029 | *** |
| Autocons_Share (% of total value) | -2.0922 | -2.1099 | 0.7635 | *** | -0.4726 | 0.2496 | * | -2.5825 | 0.9487 | *** |
| **Capital & factor constraints** | | | | | | | | | | |
| Finance_asset (FCFA10³) | 0.0102 | 0.0102 | 0.0043 | ** | 0.0023 | 0.0013 | * | 0.0125 | 0.0053 | ** |
| Human_Wildlife (FCFA10³) | -0.2179 | -0.2172 | 0.0975 | ** | -0.0488 | 0.0297 | * | -0.2660 | 0.1213 | ** |
| **HOUSEHOLDS CHARACTERISTICS** | | | | | | | | | | |
| Gender (1 = Male) | 0.5816 | 0.5802 | 0.3326 | * | 0.1300 | 0.0944 | | 0.7102 | 0.4122 | * |
| Age (continuous, in years) | 0.0301 | 0.0301 | 0.0113 | *** | 0.0067 | 0.0037 | * | 0.0368 | 0.0140 | *** |
| Ages_thr | -0.0013 | -0.0013 | 0.0006 | ** | -0.0003 | 0.0002 | | -0.0017 | 0.0008 | ** |
| Hsize (continuous) | 0.1650 | 0.1649 | 0.0369 | *** | 0.0369 | 0.0164 | ** | 0.2018 | 0.0475 | *** |
| Seniority (continuous, in years) | 0.0399 | 0.0403 | 0.0080 | *** | 0.0090 | 0.0039 | ** | 0.0494 | 0.0105 | *** |
| CommunityGroup Dummy (1 = yes) | 0.5718 | 0.5729 | 0.3161 | * | 0.1281 | 0.0896 | | 0.7010 | 0.3914 | * |
| Baka_employement (coutinuous) | 0.1584 | 0.1592 | 0.0495 | *** | 0.0356 | 0.0177 | ** | 0.1948 | 0.0622 | *** |
| **CONTEXTUAL VARIABLE** | | | | | | | | | | |
| Country (1 = Cameroun. 0 = Gabon) | 0.8565 | 0.8515 | 0.3778 | ** | 0.1786 | 0.0921 | * | 1.0300 | 0.4406 | ** |

*, ** and *** = significance level at 1% 5% 10% respectively

A CFAF10³ more monthly income (that is $1.61) from diversified strategies made of agriculture, cocoa, and forest-based activities (**ACF**), corresponds to 0.0102 more ha (102m 2) household's deforestation; with a positive spillover effect of 0.0023 ha (23m 2) within his/her neighborhood. The resulting total effect is 0.0125 ha (125m 2). Translating into dollars, using the 2014 exchange rate (CFAF 1 = $0, 0021), a $1000 more income from the ACF approximately corresponds to 4.9 more hectares own land holding with a spillover effect of 1.1 ha due to the neighborhood. That is a total effect of 6ha. Likewise, a one-unit of the monthly income, of a household head who chooses cocoa and forest (**CF**), corresponds to 0.0113 ha (113m 2) more own deforestation, with a positive spillover effect of 0.0025 ha (25m 2) within his/her neighborhood. This approximately equates to 5.4 ha additional own deforestation with a spillover effect of 1.2 ha within the neighborhood, associated to a per annum $1000 more household income. The total effect is 6.6 ha.

Household heads choosing agriculture and forest and earning CFA1000 more from their strategy, would be responsible for 0.0025 ha more deforestation from additional unit increase in the monthly income. This approximately equates to an incremental increase of own deforestation by 1.2 ha resulting from per annum $1000 increase in the household income. Households' heads choosing (**AF**) portfolio do not exert any significant spillover effect within their neighborhood.

The direct effect of increasing of households specializing in agriculture (**A**) is a significant increase by 0.0030 ha of own deforestation. The resulting significant and positive spillover effect within the neighborhood is 0.0007 ha. The resulting total effect is 0.0037 ha. This approximately equates to an incremental increase of own deforestation by 1.43 ha with a spillover effect of 0.33 ha within the neighborhood, resulting from per annum $1000 increase in the household income. That is a total effect of 1.76 ha. Households with CFA1000 more monthly income from cocoa (**C**) hold about 0.0180 ha from their own decision, with an additional 0.0040 more hectare driven by the neighborhood effect. The resulting total effect is 0.022 ha. This approximately equates to an incremental increase of own deforestation by 8.6 ha with a spillover effect of 1.9 ha within the neighborhood, resulting from per annum $1000 increase in the household income. The total effect is of 10.5 ha.

**Capital and factor constraint decision's variables**: Financial assets and the damage cost from human-elephant conflict have both significant direct and indirect effects (Table 5).

An additional unit of a loan contracted (or transfer received) by a household head leads to a marginal increase of own deforestation of 0.0102 ha with a positive spillover effect within his/her neighborhood of 0.0043 ha. That is a total effect of 0.0145 ha. This equates to an increase of own deforestation by 6.9 ha with a spillover effect of 2.04 ha within the neighborhood, resulting from per annum $1000 increase in the financial asset. That is a total effect of 8.94 ha. This indicates that increasing money transfers in favor of households living in the landscape may foster engagement in forest clearing by households.

Unlike the financial asset, the monthly cost of crops damaged by elephants exerts a negative and significant direct and indirect effect on the households' deforestation. Indeed, an additional unit of damage cost reduces both own and neighboring deforestation by 0.217 ha and 0.048 ha. The resulting total effect is 0.265 ha. This approximately equates to an incremental decrease of 103 ha and 22.8 ha respectively, resulting from per annum $1000 increase in damage cost. This result translates into two complementary effects. Firstly, a business discouragement effect that could lead to the abandonment of spaces nearby the promenade area of elephants; on the other hand, it may cause a switch from activities relating to land use to forest extraction activities that seem least risky with a possibility of increasing forest degradation.

The existence of land conflict among households (Human-Human conflict) was insignificant (see S1 Table). The third result gives the insight that there is little constraint on land

access in our case study: deforestation and agricultural expansion are not impacted by neighbor conflicts.

**Households characteristics**: There are no significant Education, marital status, and ethnicity differences in households' deforestation as shown in S1 Table. Those with significant effects include gender, age, household size, and the residence duration or seniority.

Table 5 shows that men are associated with 0.58 ha more deforestation than women without spillover effects within the neighborhood.

Deforestation increases slowly and significantly with the household head age with some threshold effect. For every year they get older, deforestation increases by 0.03 ha, with a negligible spillover effect. Larger household size induces more deforestation. Indeed, an additional member of a family increases own deforestation by 0.16 ha with a spillover effect of 0.034 ha. As pointed out by Kaimowitz and Angelsen [60], the residence duration is positively associated with forest clearing with the same level as age.

When it comes to labor, it is interesting to note that Baka employment is related to larger deforestation, while household labor duration has no influence (see S1 Table). Thus both types of labor do not seem to be substitutes. Baka labor appears to be more land-use oriented, while household labor seems not to be related to land use choices, i.e. more related to labor-intensive practices.

**Contextual variables**: Households' deforestation does not differ with "distance to market" and with "distance to the nearest protected area". In a context of low population density and a weak market, market proximity does not have a significant impact. However, as shown by Ngouhouo-Poufoun et al. [98], distance to markets influences the livelihood strategies. Thus the effect of distance indirectly passes through the livelihood strategies transmission channel. The second result indicates that public policies such as protected areas do not bring constraints on land use decisions. Moreover, we do not find evidence of leakage between protected areas and neighboring households.

Finally, both the direct and indirect effects of a country are positive and significant. Indeed, households living in Cameroon are associated with 0.85 ha more deforestation compared to those living in Gabon, with a spillover effect of 0.18 ha on proximate households in Gabon. This result suggests paying additional attention to the Cameroonian segment of the landscape. Indeed, the variation in deforestation rates between Gabon and Cameroon in a transboundary landscape highlights the urgency of strong institutions that can prevent transboundary leakage and promote the integration of conservation efforts with poverty alleviation or improvements in income and well-being for cocoa farmers in forested areas, particularly in conservation landscapes like the Tridom.

## 7 Discussion and conclusion

The aim of this study is to better understand how livelihood strategies are associated with deforestation in an open context where the agricultural sector is not yet well framed in a perspective of reducing deforestation. To that matter, this paper is a natural extension of [98], which determines the variables influencing livelihood strategies. We also develop a spatial approach in order to take into account spatial interactions between agents. Our analysis relies on an original household survey collected in the Tridom landscape.

First, when it comes to livelihoods, diversification and specialization strategies have different impacts. Strategies incorporating agricultural activities all tend to have an impact on deforestation. The corollary to this result is that only agents specializing in forest-based activities do not influence deforestation. More precisely, households with cocoa production-based activity portfolios are those making the largest impact on deforestation. Further, spillover / indirect

effects from own cocoa production-based strategies (**C, CF, ACF**) on neighboring deforestation have almost the same scale as the direct effect of the remaining strategies (**AF, A**) on own deforestation. Cocoa as a specialization strategy has the highest influence on forest cover compared to the other strategies, with a spillover effect that is almost twice as large as the direct effect caused by households mixing agriculture and forest.

Cocoa, as currently produced in the tridom landscape, is an agricultural commodity entailing a high risk of deforestation. These results bring the insight that, if development leads to households switching from small-scale agriculture to internationally traded commodities such as cocoa as a main activity, this would result in a significant increase in deforestation. As an example, one extra dollar earned in cocoa appears to be associated with a 7-times larger effect on deforestation than an extra dollar in subsistence agriculture. Cocoa cultivation can thus lead to conversion of forest to agricultural land and income from cocoa can finance other activities related to the clearing of forest to expand the household farming system. Lock and Alexander [116] show that even when combining cocoa production and sustainability the resulting production intensification attracts new farmers at the forest frontier, which ultimately leads to further deforestation. Mimicry and resulting spatial spillover effects make cocoa an inherently high deforestation risk crop under weak land governance regimes.

The current findings support previous studies on the connection between income and conservation. They contribute to the ongoing debate on whether conservation initiatives can effectively be combined with poverty alleviation in rural and forested areas of the tropics. Ruf and Goetz [117] have documented how the expansion of cocoa cultivation in the Rent Forest has been linked to deforestation. Ivory Coast serves as an illustration of how the pursuit of cocoa-related prosperity has unfortunately resulted in deforestation, despite the potential for cocoa agroforestry to play a positive role if proper conservation policies were implemented. The relationship found in this paper between income from cocoa production and deforestation has been a constant concern within the scientific community, as highlighted by Samii et al. [118]. Chiu [119] has also documented the evident link in certain circumstances.

At the moment, the Zero-deforestation policies aim at reducing the ecological footprint related to the establishment of cocoa. More attention needs then to be put on the use of cocoa income for the sustainable development of the family and the forest fringe landscape. Current efforts on REDD+, such as the cocoa forest initiative in West Africa are putting effort into structuring cocoa farmers' associations so that they contribute to the zoning of cocoa landscapes in forest fringe, but that farmers did not conduct activities that can negatively impact the local forest.

In the same vein, the share of auto-consumption is negatively related to deforestation. Here again, if economic development brings better market access and lower auto-consumption shares, this is likely to positively influence deforestation.

Second, land conflicts and distance to protected areas do not seem to influence deforestation. This result brings the insight that competing land uses is not really a matter of constraint for households, nor does it represent a source of leakage in the area. In contrast, human-wildlife conflicts do seem to have a negative impact on deforestation. Therefore, if policies are set with the aim to protect wildlife in rural areas and decrease human-wildlife interactions at the same time [86], it is a crucial matter to monitor and involve local populations in order to avoid a bump in deforestation.

Third, our paper underlines the importance of assessing deforestation factors in a spatial context. Indeed, spatial spillovers tend to be of large magnitude: indirect effects may reach up to 20% of the direct effects. This result is important, as it shows that micro-economic analysis of deforestation factors should take into account those spatial interactions, in order to have an accurate understanding of the mechanisms in place.

Fourth, labor allocation is important. The household labor duration does not seem to impact deforestation, while Baka employement is associated to higher landholding. Household labor appears to be allocated to labor-intensive activities, while Baka labor seems to be allocated to activities requiring more land and thus more deforestation. Therefore, both types of labor cannot be considered as substitutes, especially when it comes to land use. If we consider a Chayanovian approach [60], it seems that the trade-off between household labor and leisure does not influence deforestation.

Fift, factors such as gender and age have an impact on deforestation. Farming cocoa requires more physical effort and more land compared to crop growing. This explains the higher deforestation by men compared to women. These findings can be the result of the gender division of labor raised by Holden [120] as it kept female-headed households from clearing many forests.

Overall, our paper brings some insights into how development and internationally traded commodities such as cocoa may influence deforestation in such a rural landscape located in biodiversity important forest fringes of the Congo Basin. Three transmission channels are to be distinguished: an income channel, an activity portfolio channel, and a market integration channel. First, economic development comes with a larger income. We show that, except for households specializing in forest-based activities, an increase in income is related to more deforestation. Second, the portfolio of activities is likely to change with economic development, with the increased importance of cocoa and cash-cropping. This would also result in larger deforestation rates. Finally, when households have better access to markets, they tend to decrease their share of auto-consumption, which can also have a tendency to increase deforestation. It is important for development projects and policies to take those three channels into account when dealing with possible environmental adverse effects.

In the context of emerging international trade regulations on free-deforestation commodities, the question of a development model which improves living standards and the resilience of households, while preserving forests is of an urgent matter. An approach to tackle the high risk of deforestation associated with cocoa could pass through farmers by promoting a complex cocoa agroforestry system (see Sonwa et al.[27–29]) coupled with better land covers/use planning and incentives enforcement for sustainable practices. Further work should assess the challenges faced by farmers that constitute a serious bottleneck to higher yield. Indeed, producing with higher yield absorbs an important part of the effort allocated to deforestation.

## Supporting information

**S1 Fig. Diagnostic.** Diagnostic Plots for Regression Analysis.
(TIFF)

**S2 Fig. Livelihood values.** Livelihood Strategies and Per Annum Yiels/ha.
(TIF)

**S3 Fig. Correlogram for main variables.** To assess the high correlation and to detect the possible presence of multicollinearity in the data, the study applied the Pearson correlation matrix. Results of Pearson's correlation matrix indicated that the highest correlation among variables was 0.5; hence, there is no issue of multicollinearity.
(TIF)

**S1 Table. Full model.** Full Spatial Autoregressive Model with all the explanatory variables.
(ZIP)

**S2 Table. Instrumental variable.** Estimations with Instrumental variables method to control for endogeneity bias.
(ZIP)

**S3 Table. Robustness check.** Akaike's information criterion (AIC) and Bayes' information criterion (BIC): Comparison tests for models.
(ZIP)

**S1 Appendix.**
(TIFF)

**S2 Appendix.**
(TIFF)

## Acknowledgments

The Congo Basin Institute (CBI) is a partnership in international development between universities, NGOs, and private businesses formed by The University of California Los Angeles (UCLA) and the International Institute of Tropical Agriculture (IITA).

The authors thank Protet Essono, Hervé Kanna, Christianne Djole, Juvénale Nzanre, Christian Onanena, Castane Medou, Daniel Ogolong for their participation and unconditional support as regards the success of the whole households' survey in Cameroon and Gabon.

Lisen Runsten, Tanya Payne, Willy Francis Ntcheukou Nana, and Arrey Laquinez carried out the editorial work on this paper. We are very thankful to them. We are also grateful to Richard Suffo Kankeu for GIS assistance.

The authors thank the anonymous reviewers for their valuable suggestions.

## Author Contributions

**Conceptualization:** Jonas Ngouhouo-Poufoun, Philippe Delacote.

**Data curation:** Jonas Ngouhouo-Poufoun, Sabine Chaupain-Guillot.

**Formal analysis:** Jonas Ngouhouo-Poufoun, Sabine Chaupain-Guillot.

**Funding acquisition:** Jonas Ngouhouo-Poufoun, Denis Jean Sonwa, Kevin Yana Njabo.

**Investigation:** Jonas Ngouhouo-Poufoun, Denis Jean Sonwa.

**Methodology:** Jonas Ngouhouo-Poufoun, Sabine Chaupain-Guillot, Youba Ndiaye, Philippe Delacote.

**Project administration:** Jonas Ngouhouo-Poufoun, Denis Jean Sonwa, Kevin Yana Njabo, Philippe Delacote.

**Resources:** Youba Ndiaye, Philippe Delacote.

**Software:** Jonas Ngouhouo-Poufoun, Sabine Chaupain-Guillot.

**Supervision:** Denis Jean Sonwa, Kevin Yana Njabo, Philippe Delacote.

**Validation:** Jonas Ngouhouo-Poufoun, Sabine Chaupain-Guillot, Youba Ndiaye, Denis Jean Sonwa, Kevin Yana Njabo, Philippe Delacote.

**Visualization:** Jonas Ngouhouo-Poufoun.

**Writing – original draft:** Jonas Ngouhouo-Poufoun, Sabine Chaupain-Guillot, Philippe Delacote.

**Writing – review & editing:** Jonas Ngouhouo-Poufoun, Youba Ndiaye, Denis Jean Sonwa, Kevin Yana Njabo, Philippe Delacote.

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
