## [Decision Letter · Decision Letter 0]

5 Jul 2023

PONE-D-23-09897Cocoa, livelihoods and deforestation in the Congo Basin: A spatial analysisPLOS ONE

Dear Dr. Ngouhouo-Poufoun,

Thank you for submitting your manuscript to PLOS ONE. After careful consideration, we feel that it has merit but does not fully meet PLOS ONE’s publication criteria as it currently stands. Therefore, we invite you to submit a revised version of the manuscript that addresses the points raised during the review process.

We look forward to receiving your revised manuscript.

Kind regards,

Essossinam Ali, Ph.D

Academic Editor

PLOS ONE

Journal Requirements:

“1. Funding for fieldwork provided by the Norwegian Agency for Development Cooperation (NORAD), grant no: QZA-12/0882. Grants received by JNP. The funder did not play any role in the study design, data collection and analysis, decision to publish, or preparation of the manuscript. Url: https://www.norad.no/en/front/

2. We acknowledge funding for the Staff (JNP) time from the UK Research and Innovation's Global Challenges Research Fund (UKRI GCRF) through the Trade, Development and the Environment Hub project (project number ES/S008160/1). Url: https://gtr.ukri.org/projects?ref=ES%2FS008160%2F1“

“The Congo Basin Institute (CBI) is a partnership in international development between universities, NGOs, and private business formed by The University of California Los Angeles (UCLA) and the International Institute of Tropical Agriculture (IITA). The UMR BETA is supported by a grant overseen by the French National Research Agency (ANR) as part of the ”Investissements d’Avenir” program (ANR-11-LABX-0002-01, Lab of Excellence ARBRE)”

“1. Funding for fieldwork provided by the Norwegian Agency for Development Cooperation (NORAD), grant no: QZA-12/0882. Grants received by JNP. The funder did not play any role in the study design, data collection and analysis, decision to publish, or preparation of the manuscript. Url: https://www.norad.no/en/front/

2. We acknowledge funding for the Staff (JNP) time from the UK Research and Innovation's Global Challenges Research Fund (UKRI GCRF) through the Trade, Development and the Environment Hub project (project number ES/S008160/1). Url: https://gtr.ukri.org/projects?ref=ES%2FS008160%2F1“

4. We note that Figure 5 in your submission contain map images which may be copyrighted. All PLOS content is published under the Creative Commons Attribution License (CC BY 4.0), which means that the manuscript, images, and Supporting Information files will be freely available online, and any third party is permitted to access, download, copy, distribute, and use these materials in any way, even commercially, with proper attribution. For these reasons, we cannot publish previously copyrighted maps or satellite images created using proprietary data, such as Google software (Google Maps, Street View, and Earth). For more information, see our copyright guidelines: http://journals.plos.org/plosone/s/licenses-and-copyright.

 a. You may seek permission from the original copyright holder of Figure 5 to publish the content specifically under the CC BY 4.0 license. 

Additional Editor Comments:

Dear Authors,

Although the reviewers recommend a possible publication of your paper, many issues, starting from the title to references, need to be addressed. There are many words in French in the text. The article should be proofreaded before any resubmission. The theory of sustainability should be discussed in the introduction Chapter.

The following paper may strengthen the background of your paper.

Ali, E. (2021). Farm Households’ Adoption of Climate-smart Practices in Subsistence Agriculture: Evidence from Northern Togo. Environmental Management 67, 949–962; https://doi.org/10.1007/s00267-021-01436-3

Reviewers' comments:

Reviewer's Responses to Questions

**Comments to the Author**

1. Is the manuscript technically sound, and do the data support the conclusions?

Reviewer #1: Yes

Reviewer #2: Yes

2. Has the statistical analysis been performed appropriately and rigorously? 

Reviewer #1: Yes

Reviewer #2: Yes

3. Have the authors made all data underlying the findings in their manuscript fully available?

Reviewer #1: Yes

Reviewer #2: Yes

4. Is the manuscript presented in an intelligible fashion and written in standard English?

Reviewer #1: Yes

Reviewer #2: Yes

5. Review Comments to the Author

Reviewer #1: This paper analyzes the full set of potential drivers of households’ deforestation, prioritizing or distinguishing among them to inform policy makers and facilitate appropriate political decision processes to curb deforestation from smallholders’ agriculture and forest activities in the medium- and long-term perspectives. Spatial autoregressive method was used for analysis of survey data from 1035 households in the Tridom landscape in the Congo basin. The result is that households emulate the deforestation decisions of their neighbors. Moreover, a marginal increase in income from cocoa-based livelihood’s portfolio, is associated with six to seven times higher deforestation than other livelihoods’ strategies with a significant spillover effect on neighboring households’ deforestation. Increased income from cocoa-based livelihoods in open access systems can negatively affect forests. This paper proposes as a policy recommendation that an approach to tackle the high risk of deforestation associated with cocoa could pass through farmers. This is done by promoting a complex cocoa agroforest system.

Comments:

It is a very valuable and relevant topic. It contributes significantly to the development of strategies for reducing deforestation associated with cocoa. The study presents original research results and has not been published elsewhere. Also, Authors have addressed the critical concerns raised by preview reviewers.

One point that I would like to raise is the use of a spatial autoregressive model (SAR) without prior justification. For example, why is a SAR model preferred to Spatial Durbin Model (SDM) or Spatial Error-correction Model (SEM)? Bayesian and AIC information criteria could be applied in order to confirm that SAR is the best method.

Reviewer #2: This paper considers how and the pathways through which the choice of livelihood strategies influence deforestation within the Tridom landscape in the Congo Basin using deferent models of spatial analysis. The structure is appropriate and the content of the paper is very rich and contributes to the theoretical and conceptual knowledge on direct and indirect, spillover effect of various factors influencing deforestation with special focus on livelihood strategies, household factors, and capital and factor constraints decision and contextual characteristic as they found income channel, an activity portfolio channel, and a market integration channel to be crucial to addressing the challenges of deforestation. This paper really contributes to the body of literature by filling the gaps identified. The literature review section is okay and adequate as the authors considered past studies from the macro, micro and spatial perspectives to channel the cause for this paper. The theoretical and conceptual framework presented is logical, well presented, and clear. The data, spatial econometric models, and selection procedure used were clear stated and specified. Although, the discussion needs to be supported with relevant related literature which the authors are yet to do with regards to the paper. The conclusion is sharp enough and clearly presented.

Nevertheless, there are few specific comments to further improve the quality of the paper.

1. I suggest that the authors perform a through editorial work on this paper. There are several disjointed sentences and the authors needs to recast some areas for clarity.

2. I suggest that the title should read; Cocoa, livelihoods and deforestation within the Tridom landscape in the Congo Basin: A spatial analysis. Given the center focus and the study area is Tridom landscape which is in the Congo Basin.

3. Cocoa is a tree crop and not livelihood strategy but its production/cultivation. See line 3(Abstract), line 277 and 280. I suggest it should read cocoa production.

4. The following sentences in the introduction section (line 2-3; 22-25,) should be referenced. As data quoted is not original to the authors as presented.

5. Line 33 the sentence “In the Congo Basin, Cameroon, …….” From the paper, Cameroon is among the countries in the Congo Basin. The authors will need to relook at the construction of the sentence again.

6. Line 44, “about 85,45% of households ……..” Is this 80-45% or 85.45%? try and reconfirm this.

7. Line 63, “75% cocoa and plantain yield ……” this should read 75% of cocoa ….

8. Line 95, “The spatial economic procedure in presented in section 4…. This should read “is presented in section 4, ……..

9. There are areas where it is important to mention/refer to the name of author cited (see line 265, 653)

10. Line 281, “influence deforestation.” remove the space before the full-stop.

11. Line 312, “Financial asset drive households’ deforestation Walker et al. [79, 80].” Remove the name of the authors and cite with number only. Also, there are different perspective to financial asset (e.g. household financial asset, institution or factor constraints). I suggest should try and position this well in the paper as it appears loosely placed as the authors discussed it close to environmental and policy factors.

12. Line 447-449, on the average, 4.75ha of land be cleared for subsistence farming or cash-crop such as cocoa? This seems the farmers here are not smallholder, cultivating this for subsistence is questionable. Try and relook at this again.

13. In the presentation and interpretation of results section, the author should introduce a uniform currency (i.e., stick with one currency) instead of interchanging CFA with $ even though there is conversion ratio (i.e., CFAF 1 = $0, 0021) in the footnote. For instance, see line 577, 588, 601, 602, 606

14. The authors failed to discuss their results within the context of other related literatures from past scholars even though they had a very rich literature review. This is one major flop of this paper. I advised the authors to look critically at this.

15. Line 720, the authors mentioned “cacao” instead of Cocoa. If the author wants to keep it as “cacao”, this should be italicized since it is not an English name. If the botanical name is preferred, references should be made to it at the first mention or as a foot note.

16. Line 722-723, “an income channel, an activity portfolio channel, and a market integration channel”. Why italicize this key word? I think it is not necessary.

6. PLOS authors have the option to publish the peer review history of their article (what does this mean?). If published, this will include your full peer review and any attached files.

Reviewer #1: No

Reviewer #2: No

---

## [Author Response · Author response to Decision Letter 0]

5 Oct 2023

Dear Editor (updated), 

First of all, we would like to thank the editor and the reviewers for their helpful comments. We did our best to address their issues, and we hope they will find our revision satisfactory. Our detailed response below is organized into four sections. Section 1 addresses the journal requirements. Section 2 addresses the comments provided by reviewer 1. Section 3 addresses the comments from reviewer 2, and section 4 addresses the additional Editor Comments.

1 Journal requirements addressed

• Requirement 1: Please ensure that your manuscript meets PLOS ONE’s style requirements, including those for file naming for Figures:

Answer: We followed the PLOS ONE style templates to make sure that our manuscript meets PLOS ONE’s style requirements, including the file naming for Figures. 

• Requirement 2: Please state what role the funders took in the study. If the funders had no role, please state: ”The funders had no role in study design, data collection, and analysis, the decision to publish, or preparation of the manuscript.” If this statement is not correct you must amend it as needed. 

Answer: The financial disclosure was amended. We amended the statement of the role the funders took in the study. We exactly stated the following: ”The funders had no role in study design, data collection, and analysis, the decision to publish, or preparation of the manuscript”. The amended role is inserted in the cover letter.

• Requirement 3: In the Acknowledgments section of the manuscript: Please remove any funding-related text from the manuscript and let us know how you would like to update your Funding Statement. 

Answer 1: The Acknowledgments Section of the manuscript is amended. We removed the following information that relates to funding. ”The UMR BETA is supported by a grant overseen by the French National Research Agency (ANR) as part of the ”Investissements d’Avenir” program (ANR-11-LABX- 0002-01, Lab of Excellence ARBRE)”. And we inserted it in the financial disclosure. The amended acknowledgment and the amended funding disclosures are added to the cover letter. 

Answer 2: The financial disclosure is now as follows: 

 1 The Center for International Forestry Research-Global Comparative Study (CIFOR-GCS M3) has contributed to the field work with funding provided by the Norwegian Agency for Development Cooperation (NORAD), grant no: QZA-12/0882. Grants received by JNP. The funders had no role in study design, data collection, and analysis, decision to publish, or preparation of the manuscript. https://www.norad.no/en/front/. 

 2 We acknowledge funding for the Staff (JNP) time from the UK Research and Innovation’s Global Challenges Research Fund (UKRI GCRF) through the Trade, Development and the Environment Hub project (project number ES/S008160/1). The funders had no role in study design, data collection, and analysis, decision to publish, or preparation of the manuscript. https://gtr.ukri.org/projects?ref=ES%2FS008160%2F1.

 3 The UMR BETA is supported by a grant overseen by the French National Research Agency (ANR) as part of the ”Investissements d’Avenir” program (ANR-11-LABX-0002-01, Lab of Excellence ARBRE). The funders had no role in study design, data collection, and analysis, decision to publish, or preparation of the manuscript. https://anr.fr/en/.

• Requirement 4: We note that Figure 2 in your submission contains map images that may be copyrighted. For these reasons, we cannot publish previously copyrighted maps or satellite images created using proprietary data, such as Google software (Google Maps, Street View, and Earth). For more information, see our copyright guidelines: http://journals.plos.org/plosone/s/licenses-and-copyright. 

Answer: We have addressed this at page 21 and we hope we have done proper attribution, given the World Resources Institute attribution requirements. The Figure caption is as follows:

Fig 2. Livelihoods and Mean deforestation in the landscape

Source of data: Protected area, TRIDOM limit, Central Africa, Administrative limit: Country-specific land-use and administrative data come from the Forest Atlases published by the forest ministries (Ministry of Forests and Fauna in Cameroon, Ministry of Water, forests, the Sea and the Environment, in charge of the climate plan and the land use plan in Gabon, and Ministry of Forest Economy in the Republic of Congo). All these are Open Data, and were collected, digitalized, harmonized, and published by World Resources Institute (www.wri.org) in 2014, and are also available on the Forest Atlas platform and Global Forest Watch.

Cameroon Forest Atlas, the Ministry of Forest and Fauna/World Resources Institute accessed on (10/03/2023), https://cmr.forest-atlas.org/.

Republic of Congo Forest Atlas, Ministry of Forest Economy in the Republic of Congo /World Resources Institute was accessed on (10/03/2023), https://cog.forest-atlas.org/.

Gabon Forest Atlas, Ministry of Water, forests, the Sea, and the Environment, in charge of the climate plan and the land use plan /World Resources Institute, accessed on (10/03/2023), https://gab.forest-atlas.org/pages/maps

Deforestation: Data collected in the field in Gabon and Cameroun during the fieldwork carried out by the Authors in 2013 - 2014. A total of 526 plots were tracked with a Global Positioning System (GPS) to have the real area.

Household strategies: The income from livelihoods strategies in the map are calculated by the authors using data collected during the fieldwork carried out by the Authors in 2013 - 2014 in Gabon and Cameroun

2 Answer to Reviewer #1

Comment: On the justification of the use of a spatial autoregressive model (SAR): why is a SAR model preferred to Spatial Durbin Model (SDM) or Spatial Error-correction Model (SEM)? Bayesian and AIC information criteria could be applied in order to confirm that SAR is the best method. 

Answer: This comment is addressed in the manuscript in three steps. The first two steps were in the manuscript initially submitted the third is now added as a complementary answer to the comment. 

– Step 1: In our initial manuscript, we explain the selection procedure in 4.2 (lines 432-436). ”After estimating an SLM, we first tested for the existence of spatial autocorrelation using the ”Moran i” statistic on the residual of the linear model. Further, we proceeded to the Lagrange Multiplier test which helps to find the type of spatial effects that fit with our data generation process. Tables 2 and 3 in subsection 5.1 display our procedure of model specification.” 

– Step 2. We proceeded with testing the spatial dependence as part of the results in 5.1 (Lines 489-495). ”The Lagrange multiplier test presented in table 3 is used to diagnose the type of spatial dependence that governs our data generation process among the endogenous effects, i.e. spatial lag of the dependent variable (ρ /= 0) and the correlated effects or the spatial autocorrelation of the disturbance term (λ /= 0). This test suggests rejecting all the specifications that allow spatial autocorrelation in the disturbance term. Therefore, we avoid estimating the SARAR, the SEM, and the generalized nested Manski spatial model. In the following, we estimate the SAR as it fits with our data generation process.” 

– Step 3. This comment from Reviewer 1 brought us to confirm the first two steps using the Akaike information criterion (AIC) and the Bayes’ information criterion (BIC) in lines 496-501. ”Comparing the Akaike information criterion (AIC) and Bayes’ information criterion (BIC), we confirm the SAR model rather than the SDM and SEM models, cf. Appendix B.3. Indeed, the test displays lower BIC and AIC estimates for SAR compare to SEM and SDM. The relative amount of information lost by the SAR model is then lower than the amount lost by SEM and SDM models. We then prefer the SAR model with lower AIC and BIC.”

3 Answer to Reviewer#2

• General Comment 

General comment 1: The discussion needs to be supported with relevant related literature which the authors are yet to do with regards to the paper. 

Answer to General Comment 1: This general comment is the same with comment n°14. This is addressed. See lines 720-735 and lines 765-769.

• Specific comments 

* Specific Comment 1: I suggest that the authors perform a through editorial work on this paper. There are several disjointed sentences and the authors needs to recast some areas for clarity. 

Answer 1: This comment is addressed. See the revised manuscript. 

* Specific Comment 2: I suggest that the title should read; Cocoa, livelihoods and deforestation within the Tridom landscape in the Congo Basin: A spatial analysis. Given the center focus and the study area is Tridom landscape which is in the Congo Basin. 

Answer 2: The title has been change from ”Cocoa, livelihoods and deforestation in the Congo Basin: A spatial analysis” to ”Cocoa, livelihoods and deforestation within the Tridom landscape in the Congo Basin: A spatial analysis”. 

* Specific Comment 3:Cocoa is a tree crop and not a livelihood strategy but its production/ cultivation. See line 3(Abstract), lines 277 and 280. I suggest it should read cocoa production. 

Answer 3: This is addressed in the document. We now refer to cocoa production as part of livelihoods strategies, and not cocoa that is a three, see line 3(Abstract), lines 296 and 299.

* Specific Comment 4: The following sentences in the introduction section (line 2-3; 22-25,) should be referenced. As data quoted is not original to the authors as presented. 

Answer 4: This comment is addressed. Please see lines 2-7 and 36-39. 

* Specific Comment 5: Line 33 the sentence “In the Congo Basin, Cameroon, . . . . . . .” From the paper, Cameroon is among the countries in the Congo Basin. The authors will need to relook at the construction of the sentence again. 

Answer 5: This is addressed. The sentence is rephrased. Please see lines 36-39. 

* Specific Comment 6: Line 44, “about 85,45% of households . . . . . . ..” Is this 80-45% or 85.45%? try and reconfirm this. 

Answer 6: The comment is addressed. This is 85.45%. See line 47. 

* Specific Comment 7: Line 63, “75% cocoa and plantain yield . . . . . . ” this should read 75% of cocoa . . . . 

Answer 7: This is addressed. See line 66. 

* Specific Comment 8: Line 95, “The spatial economic procedure in presented in section 4. . . . This should read “is presented in section 4, . . . . . . .. 

Answer 8: This is addressed, see line 112. 

* Specific Comment 9: There are areas where it is important to mention/refer to the name of author cited (see line 265, 653). 

Answer 9: This is addressed. See lines 284 and 677. 

* Specific Comment 10: Line 281, “influence deforestation.” remove the space before the full-stop. 

Answer 10: Addressed. 

* Specific Comment 11: Line 312, “Financial asset drive households’ deforestation Walker et al. [79, 80].” Remove the name of the authors and cite with number only. Also, there are different perspective to financial asset (e.g. household financial asset, institution or factor constraints). I suggest should try and position this well in the paper as it appears loosely placed as the authors discussed it close to environmental and policy factors. 

Answer 11: Addressed, see lines 331-332, the names were removed. Regarding the comment on financial assets, We refer to financial assets as Cartas & Harutyunyan [1]. Financial assets are financial instruments or financial claims arising from contractual relationships with the basis of creditor/debtor relationships. We then exclude income as a financial asset, and keep it as one of the socioeconomic characteristics. We consider Loans, and money transfers/remittances. This is highlighted in endnote 8. 

* Specific Comment 12: Line 447-449, on average, 4.75ha of land is cleared for subsistence farming or cash-crop such as cocoa? This seems the farmers here are not smallholders, cultivating this for subsistence is questionable. Try and relook at this again.

Answer 12: Referring to Rapsomanikis [2], land holding size by smallholder varies from country to country. In Asia, farms are very small, e.g. 0.24 and 0.32 hectares in Bangladesh and VietNam respectively. In Africa, smallholder farms can be relatively larger, but only marginally. In Latin American countries, smallholder farms often tend to be over 2 hectares. In Nicaragua, the average small farm size is 5 hectares. The 4.75 ha found in our study corresponds to about 3.2 plots held by farmers, that’s 1.48ha per plot on average. 

* Specific Comment 13: In the presentation and interpretation of results section, the author should introduce a uniform currency (i.e., stick with one currency) instead of interchanging CFA with $ even though there is a conversion ratio (i.e., CFAF 1 = $0, 0021) in the footnote. For instance, see lines 577, 588, 601, 602, 606. 

Answer 13:The study is carried out in a CFA franc Zone. The analysis is carried out using the currency applicable in the countries of study. Changes brought about by CFA1000, seem difficult to appreciate as it is too small. We have chosen to translate the results in dollars to ease the understanding of the scale of change brought about by income. We suggest keeping it as it stands. 

* Specific Comment 14: The authors failed to discuss their results within the context of other related literature from past scholars even though they had a very rich literature review. This is one major flop of this paper. I advised the authors to look critically at this. 

Answer 14: This is addressed. See lines 720-735 and lines 765-769. 

* Specific Comment 15: Line 720, the authors mentioned “cacao” instead of Cocoa. If the author wants to keep it as “cacao”, this should be italicized since it is not an English name. If the botanical name is preferred, references should be made to it at the first mention or as a footnote. 

Answer 15: This is addressed. See line 771. 

* Specific Comment 16: Line 722-723, “an income channel, an activity portfolio channel, and a market integration channel”. Why italicize this keyword? I think it is not necessary. 

Answer 16: Addressed, see lines 773 & 774. 

4 Additional Editor Comments 

• Comment: The theory of sustainability should be discussed in the introduction Chapter. The following paper may strengthen the background of your paper. ”Ali, E. (2021). Farm Households’ Adoption of Climate-smart Practices in Subsistence Agriculture: Evidence from Northern Togo. Environmental Management 67, 949–962; https://doi.org/10.1007/s00267-021-01436-3”. 

• Answer: This is addressed. See Lines 83-93.

Dear Editor,

We Hope we have addressed the issues raised and that our revision is thorough. 

Please we are open to addressing any other specific or general aspect we might have forgotten. 

Thank you, Yours faithfully, 

Jonas Ngouhouo-Poufoun, Ph.D.

References 

1. Cartas MJM, Harutyunyan A. Monetary and financial statistics manual and compilation guide. International Monetary Fund; 2017. 

2. Rapsomanikis G. The economic lives of smallholder farmers: An analysis based on household data from nine countries. Food and Agriculture Organization of the United Nations, Rome. 2015;.

---

## [Editor Report · Decision Letter 1]

9 Apr 2024

Cocoa, livelihoods and deforestation within the Tridom landscape in the Congo Basin: A spatial analysis

PONE-D-23-09897R1

Dear Dr. Jonas Ngouhouo-Poufoun,

We’re pleased to inform you that your manuscript has been judged scientifically suitable for publication and will be formally accepted for publication once it meets all outstanding technical requirements.

Kind regards,

Essossinam Ali, Ph.D

Academic Editor

PLOS ONE
---

## [Editor Report · Acceptance letter]

20 May 2024

PONE-D-23-09897R1 

PLOS ONE

Dear Dr. Ngouhouo-Poufoun, 

I'm pleased to inform you that your manuscript has been deemed suitable for publication in PLOS ONE. Congratulations! Your manuscript is now being handed over to our production team.

Kind regards, 

on behalf of

Prof. Essossinam Ali 

%CORR_ED_EDITOR_ROLE%

PLOS ONE